# REAL-TIME COMPUTER VISION ON LOW-END BOARDS VIA CLUSTERING MOTION VECTORS

## ABSTRACT

In this work, we suggest computer vision methods, specifically for video tracking and map creation from video. To this end, we utilize motion vectors and clusters, which are computed very efficiently in standard video encoders, usually via dedicated hardware. We suggest a provably good tracking algorithm for clustering these vectors, by considering them as segments. For this, we utilize a definition of a *coreset* which is essentially a weighted set of points that approximates the fitting loss for every model, up to a multiplicative factor of $1 \pm \varepsilon$. Our method supports $M$-estimators that are robust to outliers, convex shapes, lines, and hyper-planes. We demonstrate the empirical contribution of our clustering method for video tracking and map creation from video, by running it on micro-computers (Le-Potato and Raspberry Pi) on synthetic and real-world videos with real-time running time.

## 1 INTRODUCTION

The goal of this work is to provide novel computer vision methods incorporating classical machine learning strategies in contrast to the current prevalent deep learning methods, with the main goal of reducing the running time and improving the robustness.

### 1.1 VIDEO TRACKING

The problem of tracking objects in RGB videos is a well-studied problem for which numerous heuristics using various approaches were proposed. A meta-survey on such approaches Zou et al. (2019) states that in recent years there were thousands of papers published on this subject. One of the very prominent approaches is utilizing neural networks. While neural networks yield unprecedented results, such improvements come at the price of training, which along with the labeling of data is rather costly. Another challenge is the cost of utilizing the results after the training, which frequently requires at least mid-level GPUs ability to achieve 30-fps (frames per second) in real-time. Moreover, recently Su et al. (2019) demonstrated the problem where even small changes in the data may "fool" the network. We note that while there were works on addressing similar problems in recent years using more sophisticated training, see for example Chen et al. (2022), it is uncertain whether more sophisticated "attacks" could cause such or similar problems to what is shown in Su et al. (2019).

### 1.2 MOTION VECTORS

Motion vectors are computed in real-time as part of existing encoders for videos, such as H.264 (Wiegand et al. (2003)), H.265 (Pereira et al. (2002)), etc. In general, those are mapping from one frame to another, with the goal of usually minimizing some loss function, such as Mean Squared Error (MSE) between the RGB values, to allow keeping only this mapping (as a vector) and the difference between the blocks.

We consider the rather simple case where the mapping is from a frame to its previous frame. For a more detailed explanation on motion vectors and their computation, see Wiegand et al. (2003).

We illustrate the motion vectors concept in the following Figure 1.

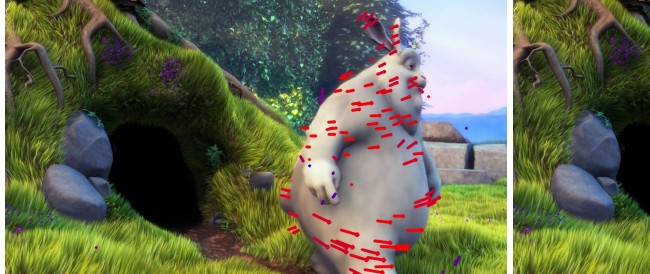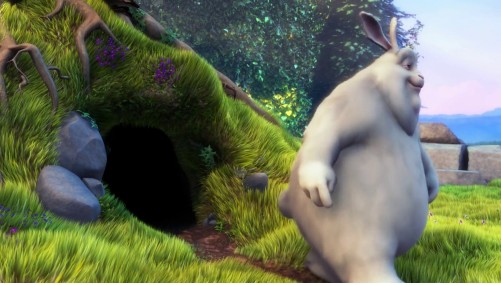

Figure 1: **Illustration of motion vectors**. The top figure is a snapshot from Roosendaal (2008) including a sub-sample of the motion vectors computed to the next frame (blue color corresponds to movement to the left and red corresponds to movement to the right), which is included below. Observe that while the most of vectors follow the movement, there is noise in the directions (either in magnitude or the direction of the moment) as evidenced by the small vector on a rock and a grass leaf in the left half of the image.

### 1.3 OUR APPROACH

We use a very different approach that does not suffer from the aforementioned drawbacks. There is no training data or training process, the computation is efficient as evidenced by our analysis and tests, and if the algorithm produces a wrong result, the reason for this faulty result can be easily traceable. We utilize motion vectors, which are computed in real-time as part of existing encoders for videos.

We will use the following definitions and notations.

**Definition 1.1.** Let $u, v$ be vectors in $\mathbb{R}^d$. A *segment* is a function $\ell : [0, 1] \to \mathbb{R}^d$ that is defined by $\ell(x) = u + vx$, for every $x \in [0, 1]$. For every pair $(p, p') \in \mathbb{R}^d \times \mathbb{R}^d$ of points, let $D(p, p') := \|p - p'\|_2$ denote the Euclidean distance between $p$ and $p'$.

We define the optimization problem as follows.

*Problem* 1 (Segment clustering). Let $S$ be a set of $n$ segments in $\mathbb{R}^d$. Let $k \geq 1$ be an integer. Let $\mathcal{Q}$ be the union over every pair $(C, w)$, where $C$ is a set of $|C| = k$ points in $\mathbb{R}^d$ and $w : C \to [0, \infty)$ is called a *weight function*. The *k-segment mean* of $S$, which we aim to find, is the set $(C, w)$ above that minimizes

$$\ell(C, w) := \sum_{s \in S} \int_0^1 \min_{c \in C} w(c) D\big(c, s(x)\big) dx. \tag{1}$$

The existence of the integral follows immediately from the continuity of the Euclidean distance.

The loss, which was denoted by $\ell$, in the function above is illustrated in Fig. 2.

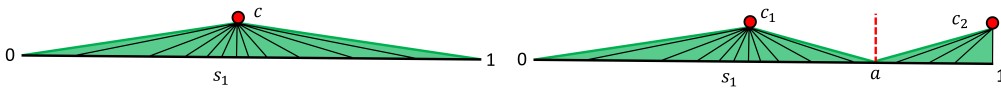

Figure 2: Visual illustration of the loss function from Problem 1. The top figure illustrates the loss function of a single center $c \in \mathbb{R}^2$ to a segment $s_1 : [0, 1] \to \mathbb{R}^2$, where the green side of the shape, which is the integral over the distances of the points on the segment to the center, is the loss of the center to the segment. The bottom figure illustrates the case where there are 2 centers $C := \{c_1, c_2\}$, as in the top figure, the size of the green shape is the loss of the centers $C$ to the segment $s_1$, but unlike to previous case, the value is the sum on two integrals where the split is at $a \in (0, 1)$. This is because the points at $\{s_1(x) \mid x \in (a, 1]\}$ are closer to $c_2$ than $c_1$, while the rest of the points (on the right) are closer to $c_2$.

The desired motion tracking is obtained by tracking the motion vectors that are assigned for each cluster. To approximate the $k$-mean defined in Problem 1 we compute a weighted set of points, which approximates the loss function, and use previous work on $k$-centers clustering by Arthur & Vassilvitskii (2007) that enables efficient approximation to the problem.

### 1.4 CORESETS

Informally, for an input set $P$ that consists of $d$-dimensional points in $\mathbb{R}^d$, a set $Q$ called *queries*, an approximation error $\epsilon \in (0, 1)$, and a loss function $\ell : P \times Q \to [0, \infty)$, a *coreset* $C$ is a data structure that approximates the loss $\ell(P, q)$ for every model $q \in Q$, up to a multiplicative factor of $1 \pm \epsilon$, in time that depends only on the size of $C$. A coreset $C$ is thus efficient if $C$ is much smaller than the original input $P$.

**Why coresets?** As stated in Denisov et al. (2023) "the main goal for constructing a coreset is to be able to compute an optimal model or its approximation much faster while losing little accuracy. Moreover, a coreset for a given problem in many cases provides a unified solution to streaming and distributed data (Braverman et al. (2016); Lu et al. (2020)), parallel computation (Feldman (2020)), handle constrained variants of the problem (Feldman & Tassa (2015)), parameter tuning (Maalouf et al. (2019)), and more (see survey in Feldman (2020)). A coreset also provides an easy solution for efficiently evaluating many different heuristics on the data, or a single heuristics with multiple different hyperparameters, in order to obtain good results without compromising time; see e.g. Maalouf et al. (2019). This is due to the two main coreset properties: (i) merge-reduce Agarwal et al. (2013); Bentley & Saxe (1980); Har-Peled & Mazumdar (2004); Indyk et al. (2014), which are usually satisfied, and (ii) the fact that a coreset approximates every model, and not just the optimal model, as in Definition 2.7."

**Coreset for segments.** Fortunately, the segment clustering problem stated in Section 2.1 has a small coreset which is a weighted subset from the input points. Often the goal is to have a coreset which is a weighted subset of the input, with few exceptions, e.g., Jubran et al. (2021) and Rosman et al. (2014), for which there was no coreset that is a weighted subset of the input, which is similar to our coreset that is a weighted set of points and not segments (that are the input).

However, since there is a significant amount of work on fitting weighted centers to weighted points, for example, see Feldman & Schulman (2012), and on the other hand we are not aware of similar works that consider the problem of fitting points to segments or even involving the non-discrete integrals, in our case a coreset that is a weighted set of points is a significantly better result that enables us to utilize the significant amount of work on fitting weighted centers to weighted points.

**Coreset for convex shapes and hyper-plane fitting.** Convex shapes can be considered as a union (of infinite size) of segments. This enables us to compute a small coreset also for the case where the input is $n$ convex shapes, as done in Algorithm 3.

Moreover, since we consider the problem of fitting weighted centers the generalization for hyper-plane fitting immediately follows, as previously mentioned at Feldman & Schulman (2012).

## 2 THEORETICAL RESULTS

In the following subsection, we state our main theoretical results, which essentially present an efficient data reduction scheme for Problem 1 that was formulated in the previous section.

### 2.1 NOTATION AND DEFINITIONS

**Notations.** Throughout this paper we assume $k, d \geq 1$ are integers and denote by $\mathbb{R}^d$ the union of $d$-dimensional real column vectors. A *weighted set* is a pair $(P, w)$ where $P$ is a finite set of points in $\mathbb{R}^d$ and $w : P \to [0, \infty)$ is a *weights function*. For simplicity, we denote $\log(x) := \log_2(x)$.

**Definition 2.1.** Let $r \geq 0$, and let $h : \mathbb{R} \to [0, \infty)$ be a non-decreasing function. We say that $h$ is *$r$-log-Lipschitz* if and only if, for every $(c, x) \in [1, \infty) \times [0, \infty)$ we have $h(cx) \leq c^r h(x)$.

**Definition 2.2.** A *symmetric-$r$* function is a function $f : \mathbb{R} \to \mathbb{R}$, such that there is $a \in \mathbb{R}$ and an $r$-log-Lipschitz function $\tilde{f} : [0, \infty) \to [0, \infty)$, that for every $x \in \mathbb{R}$ satisfies $f(x) = \tilde{f}(|x - a|)$.

In the following definitions we define lip, a global $r$-log-Lipschitz function, with $r, t, d^* \in [0, \infty)$, and $D$, a distance function; those definitions are inspired by Feldman & Schulman (2012).

**Definition 2.3 (global parameters).** Let $\text{lip} : \mathbb{R} \to \mathbb{R}$ be a symmetric-$r$ function for some $r \geq 0$, and suppose that for every $x \in [0, \infty)$ we can compute $\text{lip}(x)$ in $O(t)$ time, for some integer $t \geq 1$;

see Definition 2.1. For every weighted set $(C, w)$ of size $|C| = k$ and $p \in \mathbb{R}^d$ let $D((C, w), p) := \min_{c \in C} \text{lip}(w(c)D(c, p))$.

**Definition 2.4** (**VC-dimension**). For every $P \subset \mathbb{R}^d, r \geq 0$ and any weighted set $(C, w)$ of size $k$ let

$$\text{ball}(P, (C, w), r) := \left\{ p \in P \mid \min_{c \in C} w(c) \cdot D(c, p) \leq r \right\}.$$

Let $B := \{\text{ball}(P, (C, w), r) \mid (P, r, C, w) \in \mathcal{Q}\}$, where $\mathcal{Q}$ is the union over every $P \subset \mathbb{R}^d, r \geq 0$ and any weighted set $(C, w)$ of size $k$. Let $d^*$ denote the VC-dimension emitted by lip, which is the smallest positive integer such that for every finite $S \subset \mathbb{R}^d$ we have

$$\left| (S \cap \beta) \mid \beta \in B \right| \leq |S|^{d^*}.$$

We emphasize that the choice of the distance function $D$ dictates the appropriate values of $r, t, d^*$.

Utilizing this definition, we define an algorithm, which is stated in the following theorem, that follows from Theorem 5.1 of Feldman & Schulman (2012).

**Theorem 2.5.** *Let $k'$ denote $(k+1)^{O(k)}$. There is an algorithm* CORE-SET *that gets*

*(i). A set $P$ of points in $\mathbb{R}^d$. (ii). An integer $k \geq 1$. (iii). Input parameters $\epsilon, \delta \in (0, 1/10)$,*

*Such that its output $(S, w) :=$* CORE-SET$(P, k, \epsilon, \delta)$ *is a weighted set satisfying Claims (i)–(iii) as follows:*

*(i) With probability at least $1 - \delta$, for every $(C, w')$, a weighted set of size $k$, we have*

$$\left| \sum_{p \in P} D((C, w'), p) - \sum_{s \in S} \left( w(s)D((C, w'), p) \right) \right| \leq \epsilon \cdot \sum_{p \in P} D((C, w'), p).$$

*(ii) The size of the set $S$ is in $\dfrac{k' \cdot \log(n)^2}{\epsilon^2} \cdot O\left( d^* + \log\left( \dfrac{1}{\delta} \right) \right)$.*

*(iii) The computation time of the call to* CORE-SET$(P, k, \epsilon, \delta)$ *is in*

$$ntk' + tk' \log(n) \cdot \log \left( \log(n)/\delta \right)^2 + \frac{k' \log(n)^3}{\epsilon^2} \cdot \left( d^* + \log\left( \frac{1}{\delta} \right) \right).$$

**Definition 2.6** (Loss function). Let $\ell : [0, 1] \to \mathbb{R}^d$ be a segment; see Definition 1.1. We define the *fitting loss* of a weighed set $(C, w)$ of size $k$, as

$$\text{loss}(\ell, (C, w)) = \int_0^1 D((C, w), \ell(x)) dx.$$

Given a finite set $L$ of segments and a weighed set $(C, w)$ of size $|C| = k$, we define *the loss of fitting $(C, w)$ to $L$* as

$$\text{loss}(L, (C, w)) = \sum_{\ell \in L} \text{loss}(\ell, (C, w)). \tag{2}$$

Given such a set $L$, the goal is to recover a weighed set $(C, w)$ of size $k$ that minimizes $\text{loss}(\ell, (C, w))$.

**Definition 2.7** (($\epsilon, k$)-coreset). Let $\ell$ be a segment, and let $\epsilon > 0$ be an error parameter; see Definition 1.1. A weighed set $(S, w)$ is an ($\epsilon, k$)-*coreset* for $\ell$ if for every weighted set $Q := (C, w)$ of size $|C| = k$ we have

$$\left| \text{loss}(\ell, Q) - \sum_{p \in S} w(p) \cdot D(Q, p) \right| \leq \epsilon \cdot \text{loss}(\ell, Q).$$

More generally a weighed set $(S, w)$ is an ($\epsilon, k$)-*coreset* for a set $L$ of segments if for every weighted set $Q := (C, w)$ of size $|C| = k$ we have

$$\left| \text{loss}(L, Q) - \sum_{p \in S} w(p) \cdot D(Q, p) \right| \leq \epsilon \cdot \text{loss}(L, Q).$$

## 2.2 ALGORITHMS

In the following algorithm, we show a simple, yet robust, deterministic coreset scheme for the problem in Definition 2.6 (specifically Eq. equation 2) for the case of one segment ($n = 1$).

---

**Algorithm 1:** SEG-CORESET$(\ell, k, \epsilon)$; see Lemma 2.8.

---

**Input** : A segment $\ell : [0, 1] \to \mathbb{R}^d$, an integer $k \geq 1$, and error $\epsilon \in (0, 1/10]$; see Def. 1.1.
**Output:** A weighed set $(S, w)$, which, with probability at least $1 - \delta$, is an $(\epsilon, k)$-coreset of $\ell$;
        see Definition 2.7.

1 $\epsilon' := \left\lceil \frac{4k \cdot (20k)^{r+1}}{\epsilon} / \right\rceil$    `// r is as defined in Definition 2.3.`

2 $S := \{\ell(i/\epsilon') \mid i \in \{0, \cdots, \epsilon'\}\}$.

3 Let $w : S \to \{1/\epsilon'\}$, i.e., the function that maps every $p \in S$ to $w(p) := 1/\epsilon'$.

4 **return** $(S, w)$.

---

The main novelty of Algorithm 1 lies in bounding the contribution to the sum of each point comprising the segment, which is constant for all the points on the segment. Hence, surprisingly enough, a uniform sample could have been used as an efficient sampling at the price of introducing failure probability and larger coreset size, but instead, we use a deterministic coreset construction by generalizing previous work from Rosman et al. (2014).

Note that $r$ in Algorithm 1 depends on the distance function $D$ as stated in Definition 2.3, e.g., the value of $r$ for absolute error is 1, but for MSE it is 2.

The following lemma states the desired properties of Algorithm 1; see Lemma F.4 for its proof.

**Lemma 2.8.** *Let $\ell : [0, 1] \to \mathbb{R}^d$ be a segment, and let $\epsilon \in (0, 1/10]$; see Definition 1.1. Let $(S, w)$ be the output of a call to* SEG-CORESET$(\ell, k, \epsilon)$; *see Algorithm 1. Then $(S, w)$ is an $(\epsilon, k)$-coreset for $\ell$; see Definition 2.7.*

In the following algorithm, we utilize Algorithm 1 to turn every segment into a set of points, and then assign the union of all the points to the compression scheme in Feldman & Schulman (2012).

**Overview of Algorithm 2.** The input to the algorithm is a set $L$ of $n$ segments (see Definition 1.1), an integer $k \geq 1$, and input parameters $(\epsilon, \delta) \in (0, 1/10]$. The output is a weighted set $(P', w')$ that with probability at least $1 - \delta$, is a $(\epsilon, k)$-coreset for $L$; see Definition 2.7. This is by essentially applying Algorithm 1 to sample points from every segment, which yields a weighed set that can be used as an $\epsilon$-coreset for $L$. Then, since we obtain the problem of fitting $k$ weighed points to points with equal weight, we plug the union of the outputs of Algorithm 1 as an input to CORE-SET (see Theorem 2.5), which allows us to further reduce the size of the coreset.

---

**Algorithm 2:** CORESET$(L, k, \epsilon, \delta)$; see Theorem 2.9.

---

**Input** : A finite set $L$ of segments, an integer $k \geq 1$, and input parameters $\epsilon, \delta \in (0, 1/10]$.
**Output:** A weighted set $(P, w)$, which, with probability at least $1 - \delta$, is an $(\epsilon, k)$-coreset of $L$;
      see Definition 2.7.

1 For every $\ell \in L$ let $(P'_\ell, w_\ell) :=$ SEG-CORESET$(\ell, k, \epsilon/2)$ `// see Algorithm 1, the`
       `division by 2 is to account for the output being further`
       `reduced.`

2 $P' := \bigcup_{\ell \in L} P'_\ell$

3 $(P, w') :=$ CORE-SET$(P', k, \epsilon/4, \delta)$ `// see Theorem 2.5.`

4 $\epsilon' := \left\lceil \frac{8k \cdot (20k)^{r+1}}{\epsilon} \right\rceil$    `// r is as defined in Definition 2.3.`

5 Set $w(p) := w'(p)/\epsilon'$ for every $p \in P'$.

6 **return** $(P, w)$.

---

The following theorem states the desired properties of Algorithm 2; see Theorem G.1 for its proof.

**Theorem 2.9.** *Let $L$ be a set of $n$ segments, and let $\epsilon, \delta \in (0, 1/10]$. Let $(P, w)$ be the output of a call to* CORESET$(L, k, \epsilon, \delta)$; *see Algorithm 2. Then Claims (i)–(iii) hold as follows:*

  *(i) With probability at least $1 - \delta$, we have that $(P, w)$ is an $(\epsilon, k)$-coreset of $L$; see Definition 2.7.*

  *(ii)* $|P| \in \dfrac{k' \cdot \log^2 m}{\epsilon^2} \cdot O\left(d^* + \log\left(\dfrac{1}{\delta}\right)\right)$, *where* $m = \dfrac{8kn \cdot (20k)^{r+1}}{\epsilon}$ *and* $k' \in (k+1)^{O(k)}$.

*(iii) The $(\epsilon, k)$-coreset $(P, w)$ can be computed in order of*

$$mtk' + tk' \log(m) \cdot \log^2\left(\log(m)/\delta\right) + \frac{k' \log^3 m}{\epsilon^2} \cdot \left(d^* + \log\left(\frac{1}{\delta}\right)\right)$$

  *time, where $m = 8kn \cdot (20k)^{r+1}/\epsilon$ and $k' \in (k+1)^{O(k)}$.*

The structure of Algorithm 2 can be illustrated as follows (See Figure 3); note that we have substituted CORE-SET in Definition 2.3 by Bachem et al. (2018) for easier implementation.



Figure 3: Illustration of Algorithm 2. (Top): The input, a set of $n = 100$ segments on the plane, which is the same one sampled in Figure 1. (Middle): The union of the outputs of the calls to SEG-CORESET computed at Line 1 in the call to Algorithm 2 is the red dots (that all have the same weight) and the black lines are the input segments. (Bottom): The final coreset returned by Algorithm 2, which is obtained by further sampling from the outputs of SEG-CORESET via Bachem et al. (2018). Here, the size of each dot is proportional to its weight.

**Generalization to multi-dimensional shapes.** Since we have considered fitting centers to segments, there is a natural generalization to different shapes. The simplicity of segments allows us to generalize the data reduction for convex shapes, as done in Section B of the appendix due to space limitations.

## 3 EMPIRICAL EVALUATION; VIDEO TRACING

For simplicity, in the experimental results, we focus on the common sum-of-squared distances between objects, which corresponds to Gaussian noise, as explained in Guo et al. (2011).

**Tracing method:** The $k$-means problem entails, given a non-empty set $P \subset \mathbb{R}^d$, to find (or at least approximate) the set $C$ of $|C| = k$ points in $\mathbb{R}^d$ that minimize $\sum_{p \in P} \min_{c \in C} D(p, c)$. An efficient provable approximation for this is given at Arthur & Vassilvitskii (2007); implemented in Bradski (2000).

We implemented a video tracking method as follows, for each 10 consecutive frames of the video:

**(i).** Add for each motion vector its degree to $(0, 1)$ and $(1, 0)$, scaled such that 180 degrees corresponds to the largest end point of all the vectors; thus we have 4-dimensional segments.
**(ii).** If there are above 1000 vectors sample uniformly (with no repetitions) 1000 vectors.
**(iii).** For each 4 dimensional segment left apply the coreset at Algorithm 1, calibrated such that the coreset size for each segment is 10.
**(iv).** Compute the $k$-means, for $k = 2$, for the points obtained; done via Arthur & Vassilvitskii (2007).
**(v).** Track the mean starting position and end position of the motion vectors corresponding to the largest cluster obtained (in terms of the number of points assigned to it).

**Goal:** In those tests, we aim to demonstrate that utilizing $k$-means clustering on the coreset computed in Algorithm 1 we can in real-time and on standard hardware track the movement of objects in a video.

We emphasize that since the computation utilizes only the motion vectors, and not the RGB part of the image, this method also allows privacy preservation to some degree while providing real-time object tracking.

**Hardware:** All the experiments in this section, unless stated otherwise, were run on a standard Laptop with an Intel Core i3-1115G4, and 16GB of RAM. The video was streamed in real-time from a file, we have utilized clips at 720x1280p resolution from include Roosendaal (2008); video provided in the supplementary material.

**Software:** We implemented our algorithms in python 3.8 Van Rossum & Drake (2009) utilizing Bradski (2000), Bommes et al. (2020), Thakur et al. (2022), and Harris et al. (2020).

**Executable:** For easy validation of the results, we include an executable implementation file with a GUI interface, which was implemented via Schimansky, in the following anonymous GitHub `https://anonymous.4open.science/r/Real_time_cv_via_motion_vectors`. The execution file was built for utilization on Ubuntu 22.04 Sobell (2015) (with current Intel CPUs), and it is recommended to run the executable in a virtual box or other sandboxing measures as per the recommendation of the conference. The source code is provided in Code (2024).

## 3.1 BIG BUCK BUNNY TEST

In this test, we have used a clip of the Big Buck Bunny video Roosendaal (2008) that contains 400 frames in the resolution of 720x1280p. We have chosen this video due to its prevalence in the video tracking community, and being licensed under the Creative Commons Attribution 3.0 license, which entails "you can freely reuse and distribute this content, also commercially, for as long you provide a proper attribution"; cited from the site of the project at `https://peach.blender.org/about`. We have chosen this part of the video since it contains a bunny walking across the stationary background, thus the tracking can be validated rather simply in a visual sense.

In what follows we provide a few examples from the clip, the entire clip of the tracking result is provided in the supplementary material.

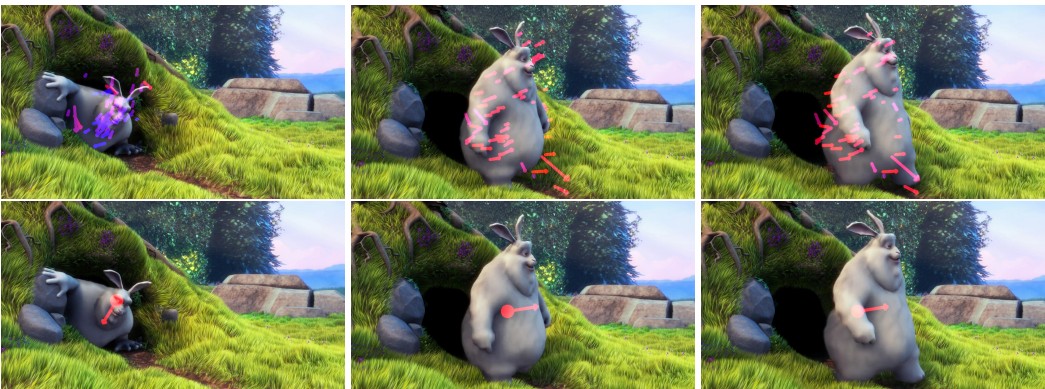

Figure 4: A subset of the results for the experiment at Section 3.1. The top row is the largest cluster of the motion vectors computed and the bottom row is the center of this cluster and mean direction. The left column demonstrates a section of the video where the bunny rises from the cave. The middle and right column demonstrate a section of the video where the bunny walks to the left and are taken with a small time difference in the video.

As can be seen in the top images the movement direction is to the opposite direction from the movement of the bunny. This occurred due to the bunny entering the field of view, and thus the closest (in color) parts from the previous frame are the parts already seen, and thus the motion vectors pointed in the opposite direction from the actual movement. Unfortunately by only looking at the motion vectors we cannot remove this problem when objects enter or disappear from the field of view. Nonetheless, observe the point in the frames, which aims to represent the center of the moving object is the center of the bunny, and thus there is no visually evident problem in the center of the cluster.

On the other hand, as can be seen in the bottom images, when the bunny has entirely entered the field of view and started walking, we obtain visually logical tracking with the predicted movement following the movement of the bunny and the cluster being the center of the object.

**Running time:** To provide context for the computation time we note that the entire running time (including decoding the video and saving the resulting tracking video) was $1.44$ seconds, of which the decoding the video and saving the resulting tracking video took $1.16$ seconds, i.e., our tracking algorithm took only $0.28$ seconds. Since there are $400$ frames in the clip, the tracking algorithm processed above $1,400$ fps, and even including decoding the video and saving the resulting tracking video we obtain $278$ fps.

Note that we have used the executable provided, with the option to output only the tracking mask, i.e., not overlay the mask, since it significantly increased the time.

To put our running time into perspective, the running time of YOLOv8 Terven & Cordova-Esparza (2023), which is an improvement over YOLOv5 Jocher et al. (2021), over the same clip is $33.4$ seconds, which entails a processing rate of 12 fps, output attached at the supplementary material.

## 3.2 Re-running for Low-end board:

We have rerun the previous test (Section 3.1) for Libre computes AML-S905X-CC (also known as Le Potato) https://libre.computer/products/aml-s905x-cc/, which is a small single-board computer similar to Raspberry Pi Upton & Halfacree (2016); we have used the official Raspberry Pi OS distributed for Le Potato. Due to missing support for Arm architecture, we extracted the motion vectors beforehand and transferred them as a Numpy array.

In this, we aim to demonstrate that our methods can support extremely low-end systems in reasonable real-time. We have obtained essentially the same results, as can be expected since the only sources for noise are ties broken arbitrarily in the clustering and noise from the samples we take.

The entire running time (excluding innit, but including decoding the video and saving the resulting tracking video) was $17.36$ seconds, of which decoding the video and saving the resulting tracking video took $13.13$ seconds, i.e., our tracking algorithm took only $4.23$ seconds. Since there are $400$ frames in the clip, the tracking algorithm processed above $94$ fps, and even including decoding the video and saving the resulting tracking video we obtain $23$ fps.

## 4 Empirical evaluation; 3D map creation

In this test, we aim to construct a 3D map from a video of a drone rotating in space.

For a fair comparison, all the computations were done on Raspberry-Pi zero Upton & Halfacree (2016) on the drone, with the map created in a time of flight, in the sense that each method gets frames in its processing rate, via PiCamera; maps created from the videos are added in the supplementary material (due to anonymity constraints we refer from submitting the original videos).

For our method we have utilized the clustering method to preemptively remove outliers from the motion vectors of each vector as follows:

**(i).** Add for each motion vector its degree to $(0, 1)$ and $(1, 0)$, scaled such that $180$ degrees corresponds to the largest end point of all the vectors; thus we have 4-dimensional segments.
**(ii).** If there are above 1000 vectors sample uniformly (with no repetitions) 1000 vectors.
**(iii).** For each $4$ dimensional segment left apply the coreset at Algorithm 1.
**(iv).** Compute the $k$-means via Arthur & Vassilvitskii (2007), for $k = 10$, over the points obtained.
**(v).** From each cluster, remove the motion vectors with a distance of more than the mean distance from the cluster with s.t.d. of distances from the cluster; assumed to be outliers.

**Estimation methods:** In our test, we consider the following map creation methods, where both are given two videos of the drone completing a 360-degree rotation at different heights.

**(i).** The classic ORB slam pipeline, as explained at Mur-Artal et al. (2015).
**(ii).** Our method utilizing the motion vectors in the ORB slam pipeline, i.e., for each frame take the start of each motion vector as a feature and map this feature to the end of the motion vector.

Note that for a fair comparison, we have utilized the gyroscope on the drone for rotation estimation, and run the processing on 4-simultaneous threads.

**Running time:** The running time of the ORB-slam pipeline was 36.4 seconds, and our running time was 21.6 of which over 80 percent was taken by extracting the motion vectors; this can be reduced almost entirely via utilization of the built-in hardware, which we have not managed to do with the current API. Therefore, since each of the two videos contained approximately 120 frames we obtained 6.6 fps for ORB-slam and 11.1 fps for our method and excluding motion vector extraction a speed of over 40 fps.

**Demonstration:** In what follows we provide our maps created from the videos, along with a demonstration of the drone utilizing the map to exit the lab room (was not organized to simulate a real-life scenario).

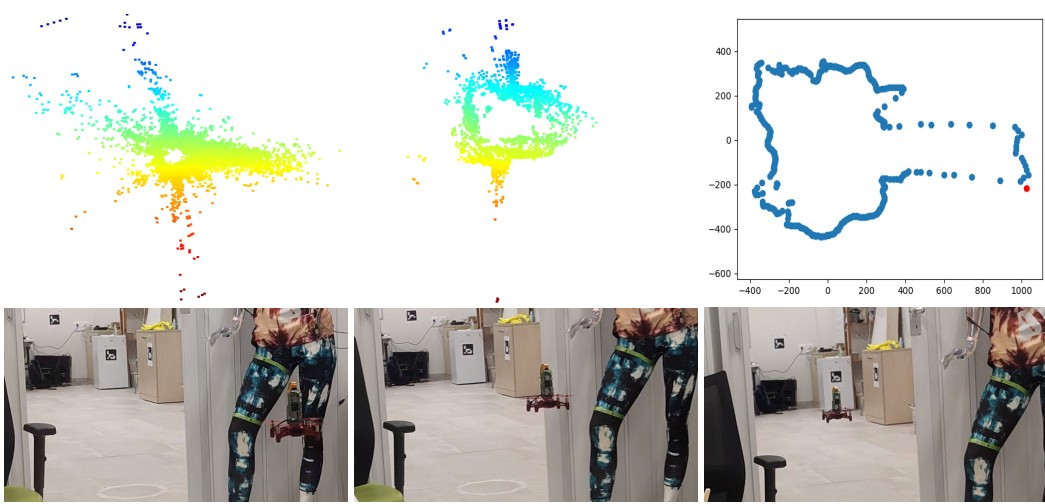

Figure 5: A subset of our results for the experiment at Section 4. The top row demonstrates the map created, with the right figure being the original map created, the middle being the map after normalization (by quantization), and the left map being a projection to 2D of the middle map (by deleting the "height" axis) along with the exit point found on it (the red dot). The bottom row demonstrates the exit of the drone utilizing the map, with the left image being the start of the exit, the middle being in the middle of the exit process, and the right being when the drone exits the lab.

## 5   DISCUSSION

Our result can be seen as a generalization of previous works, whose subset is given as follows.

**Discrete integrals.** Given $f : \mathbb{R} \to \mathbb{R}$ and integer $n \geq 1$ we refer to the value of $\sum_{i=1}^{n} f(i/n)$ as a *discrete integral* of $f$ over $[0, 1]$ for $n$. At Har-Peled (2006) it was proven that a constant weighed subset of $[0, 1]$, whose size is in $O\left(\log(n)/\epsilon^2\right)$, approximates the discrete integral up to factor of $(1 \pm \epsilon$ of every $f$ in some subset of functions over $[0, 1]$ for $n$.

**Continues integrals.** Our result can be seen as a generalization of Har-Peled (2006) to the case of the classic integral, i.e., we generalize the result from $\sum_{i=1}^{n} f(i/n)$ to $\int_0^1 f(x)dx$; note that we support a different family of functions. To the best of our knowledge, this is the first such provable result.

**Riemann sums.** Our result is related to Riemann sums, but note that the claim in the original work Riemann (1868) is an approximation for a number of samples approaching infinity and not a hard bound as we provide at Theorem 2.9.

**Feldman & Schulman (2012).** Our result at Theorem 2.9 can also be considered as an improvement to Feldman & Schulman (2012), where the input is an infinite set of points that lays on segments; note that we utilize Feldman & Schulman (2012) to further reduce our coreset's size.

Due to space limitations, we put the future work and conclusion in the appendix.

ETHICS STATEMENT

To the best of our knowledge there are no ethical concerns, all the experiments used either video filmed by the authors in a lab not including any subjects or publicly available data that "you can freely reuse and distribute this content, also commercially, for as long you provide a proper attribution"; cited from the official site for the project `https://peach.blender.org/about`.

REPRODUCIBILITY STATEMENT

As mentioned at Code (2024), we commit to publish the open source code for all the experiments in the paper upon acceptance or reviewer request. Beside this, we also include the results for all the experiments along with an executable for the experiment in Section 3.1.

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

# Appendix

## Table of Contents

# A   FUTURE WORK AND CONCLUSION

While we have not analyzed the variants of the problem where there is an additional constraint that each segment should be assigned to a single center, we believe that this would follow from incorporating our result with Section 15.2 of Feldman & Langberg (2011). We have not considered this direction since our method is based on reducing the segments to a set of points and applying the classic $k$-means clustering, which can be computed efficiently via Arthur & Vassilvitskii (2007), and even after compression this variation on Problem 1 yields a complex set clustering problem.

Our results support loss functions where the integral in equation 1 at Problem 1 is not necessarily elementary, and as such direct minimization seems to us unfeasible.

In Feldman et al. (2017) it was proven that there is a private coreset for $k$-means. Hence, we believe that due to the deterministic reduction to $k$-means in Algorithm 2, i.e., the construction of $P'$, it follows that substituting the call to Theorem 2.5 by a call to corresponding coreset from Feldman et al. (2017) would yield a private coreset for the problem in Problem 1.

# B   GENERALIZATION TO MULTI-DIMENSIONAL SHAPES

In this section we demonstrate how Algorithm 2 can be generalized to support multi-dimensional shapes, such as spheres or hyper-cubes.

This requires the following definitions.

**Definition B.1** (loss-function). Let $\mathcal{A} \subset \mathbb{R}^d$ be a convex set. We define the loss of fitting every weighed set $Q := (C, w)$ of size $|C| = k$, as

$$\text{loss}(\mathcal{A}, Q) := \int_{p \in C} D(Q, p) dV,$$

where the integral is not assumed to be elementary; $dV$ is the volume element. Given a finite set $\mathcal{K}$ of convex sets, and a weighed set $Q := (C, w)$ of size $|C| = k$, we define the loss of fitting $Q$ to $\mathcal{K}$ as

$$\text{loss}(\mathcal{K}, Q) := \sum_{\mathcal{A} \in \mathcal{K}} \text{loss}(\mathcal{A}, Q). \tag{3}$$

Given such a set $\mathcal{K}$, the goal is to recover a weighed set $Q := (C, w)$ of size $|C| = k$, that minimizes $\text{loss}(\mathcal{K}, Q)$.

**Definition B.2** (($\epsilon, k$)-coreset). Let $\mathcal{K}$ be a set of $n$ convex sets in $\mathbb{R}^d$, and let $\epsilon > 0$ be an error parameter; see Definition 1.1. A weighed set $(S, w)$ is a ($\epsilon, k$)-*coreset* for $C$ if for every weighted set $Q := (C, w)$ of size $|C| = k$, we have

$$\left| \text{loss}(\mathcal{K}, Q) - \sum_{p \in S} w(p) \cdot D(Q, p) \right| \leq \epsilon \cdot \text{loss}(\mathcal{K}, Q).$$

**Definition B.3** (well-bounded set). A convex set $\mathcal{A} \subset \mathbb{R}^d$ is *well-bounded* if there is an "oracle membership" function $\psi : \mathbb{R}^d \to \{0, 1\}$ that maps every $p \in \mathcal{A}$ to 1 and $p \in \mathbb{R}^d \setminus \mathcal{A}$ to 0, where the output can be computed in $O(d)$ time. Moreover, the smallest (by volume) non-axis parallel bounding box of $C$ is given and has non-zero volume.

This definition aims to define "not-over complex" shapes, and contains hyper-rectangles, spheres, ellipsoids.

As in Algorithm 2, the intermediate sampling at Algorithm 3 is a point set of size larger than the original input size, which is then reduced to a point set of a size that is poly-logarithmic in the size of the input set.

We emphasize that due to the random sample that is picked at Lines 5–7 of Algorithm 3 it is not guaranteed to terminate, but only has expected running time, i.e., we only prove the expected time complexity and not the worst case complexity.

The following theorem states the desired properties of Algorithm 3; see Theorem H.5 for its proof.

**Theorem B.4.** *Let $\mathcal{C}$ be a set of $n$ well-bounded convex shapes; see Definition B.3. Put $\epsilon, \delta \in (0, 1/10]$. Let $(P, w)$ be the output of a call to CONVEX-CORESET$(\mathcal{C}, k, \epsilon, \delta)$; see Algorithm 3. Let $\lambda$ as computed at Line 1 at the call to Algorithm 3. Let $n' = \lambda \cdot n$. Then there is $(k' \in k + 1)^{O(k)}$ such that Claims (i)–(iii) hold as follows:*

*(i) With probability at least $1 - \delta$ we have that $(P, w)$ is an $(\epsilon, k)$-coreset of $\mathcal{C}$; see Definition B.2.*

*(ii) The size of the set $S$ is in*

$$\frac{k' \cdot \log^2 n'}{\epsilon^2} \cdot O\left(d^* + \log\left(\frac{1}{\delta}\right)\right).$$

*(iii) The pair $(P, w)$ can be computed in*

$$n' t k' d^2 + t k' \log(n') \cdot \log^2\left(\log(n')/\delta\right) + \frac{k' \log^3 n'}{\epsilon^2} \cdot \left(d^* + \log\left(\frac{1}{\delta}\right)\right)$$

*expected time.*

---

**Algorithm 3:** CONVEX-CORESET$(\mathcal{K}, k, \epsilon, \delta)$; see Theorem B.4.

---

**Input** : A set $\mathcal{K}$ of $n$ well-bounded convex shapes, an integer $k \geq 1$, and $\epsilon, \delta \in (0, 1/10]$.
**Output:** A weighted set $(P, w)$, which, with probability at least $1 - \delta$ is an $(\epsilon, k)$-coreset for $\mathcal{K}$; see Definition B.2.

1 Set $t := (20k)^{d(r+1)}$.
2 Set $\lambda := \dfrac{c^* d^* (t+1)}{\epsilon^2} \left(k \log_2(t+1) + \log\left(\dfrac{2}{\delta}\right)\right)$, where $c^* \geq 1$ is a constant that can be
   determined from the proof of Theorem B.4.
3 **for** *every $C \in \mathcal{C}$* **do**
4 $\quad$ Let $B_C \subset \mathbb{R}^d$ be the smallest (by volume) non-axis parallel bounding box of $C$.
5 $\quad$ $S_C := \emptyset$
6 $\quad$ **while** $|S_C| < \lambda$ **do**
7 $\quad\quad$ Let $p \in B_C$ be sampled uniformly at random from $B_C$.
8 $\quad\quad$ **if** $p \in C$ **then**
9 $\quad\quad\quad$ $S_C := S_C \cup p$
10 $S := \bigcup_{C \in \mathcal{C}} S_C$.
11 $(P, w') := $ CORE-SET$(S, k, \epsilon/4, \delta/2)$; see Definition 2.3.
12 Set $w(p) := \lambda \cdot w'(p)$ for every $p \in P'$.
13 **return** $(P, w)$.

---

## C    BOUNDING THE SENSITIVITY OF ONE $r$-LIFSHITZ FUNCTION:

In this section, we bound the sensitivity of a single $r$-Lifshitz function, which, in the following section, we would expand to be a sensitivity bound for the minimum over $r$-symmetric functions. To simplify the following lemma we prove the following propositions.

**Proposition C.1.** *For every $r \geq 0$, and an integer $n \geq 2$ we have $\sum_{i=1}^{n} i^r \geq (n/2)^{r+1}$.*

*Proof.* Let $r \geq 0$, and let integer $n \geq 2$ be an integer. By sum properties we have

$$\sum_{i=1}^{n} i^r \geq \sum_{i=\lceil n/2 \rceil}^{n} i^r \geq \lceil n/2 \rceil \cdot \left(n - \lfloor n/2 \rfloor\right)^r = \lceil n/2 \rceil^{r+1} \geq (n/2)^{r+1},$$

where the second inequality is by observing that the summation is over $\{\lceil n/2 \rceil, \cdots, n\}$. $\qquad \square$

**Proposition C.2.** *Let $r \geq 0$, and $f$ be an $r$-log-Lipschitz function; see Definition 2.1. For every $x, x' > 0$ we have*

$$f(x) \cdot \min \left\{ 1, \left( \frac{x'}{x} \right)^r \right\} \leq f(x'). \tag{4}$$

*Proof.* If $(x/x') \geq 1$ we have

$$f(x) \cdot \left( \frac{x'}{x} \right)^r \leq f(x'),$$

which follows from plugging $c = (x/x'), x = x'$ in the definition of $r$-log Lipschitz functions, and dividing both sides by $(x/x')^r$. If $(x/x') \leq 1$, we have $x \leq x'$. By the definition of $r$-log Lipschitz functions we have that $f$ is non-decreasing. Hence, $f(x) \leq f(x')$. Therefore equation 4 holds. $\square$

**Lemma C.3.** *Let $n \geq 2$ be an integer. Let $r \geq 0$, and let $F$ be the set of all the $r$-log-Lipschitz functions; see Definition 2.1. For every $x \in \{1, \cdots, n\}$ we have that*

$$\max_{f \in F} \frac{f(x)}{\displaystyle\sum_{x'=1}^{n} f(x')} \leq \frac{2^{r+2}}{n}.$$

*Proof.* Let $f \in F$. By Proposition C.2, for every $x, x' \in \{1, \cdots, n\}$ we have

$$f(x) \cdot \min \left\{ 1, \left( \frac{x'}{x} \right)^r \right\} \leq f(x'). \tag{5}$$

Hence, for every $x \in \{1, \cdots, n\}$ and any $f \in F$ we have

$$\frac{f(x)}{\displaystyle\sum_{x'=1}^{n} f(x')} \leq \frac{f(x)}{\displaystyle\sum_{x'=1}^{n} \left( f(x) \cdot \min \left( 1, \left( \frac{x'}{x} \right)^r \right) \right)} \tag{6}$$

$$= \frac{1}{\displaystyle\sum_{x'=1}^{n} \min \left( 1, \left( \frac{x'}{x} \right)^r \right)} \tag{7}$$

$$= \frac{1}{\displaystyle\sum_{x'=x+1}^{n} (1) + \sum_{x'=1}^{x} \left( \frac{x'}{x} \right)^r} \tag{8}$$

$$\leq \max \left( \frac{1}{\displaystyle\sum_{x'=x+1}^{n} (1)}, \frac{x^r}{\displaystyle\sum_{x'=1}^{x} (x'^r)} \right). \tag{9}$$

where Equality equation 6 is by plugging equation 5, Equality equation 7 is by dividing both sides by $f(x)$, Equality equation 8 and equation 9 are by reorganizing the expression. If $x \leq n/2$ by Equations equation 6 to equation 9 we have

$$\frac{f(x)}{\displaystyle\sum_{x'=1}^{n} f(x')} \leq \frac{2}{n}. \tag{10}$$

If $x \geq n/2$, by equation 6 to equation 9 we have

$$\frac{f(x)}{\sum\limits_{x'=1}^{n} f(x')} \leq \frac{x^r}{\sum\limits_{x'=1}^{x} (x'^r)} \tag{11}$$

$$\leq \frac{2^{r+1} x^r}{x^{r+1}} \tag{12}$$

$$= \frac{2^{r+1}}{x} \tag{13}$$

$$\leq \frac{2^{r+2}}{n}, \tag{14}$$

where Equation 11 is by the result above, Equation 12 is since $x \geq n/2$ by the definition of the case and plugging $n = n/2$ in Proposition C.1, Equation 13 is by reorganizing the expression, Equation 14 is since $x \geq n/2$ (the definition of the case). Combining Equation 10 and Equations 11 to 14 proves the lemma. □

The main idea in the previous lemma can be demonstrated using the following figures.

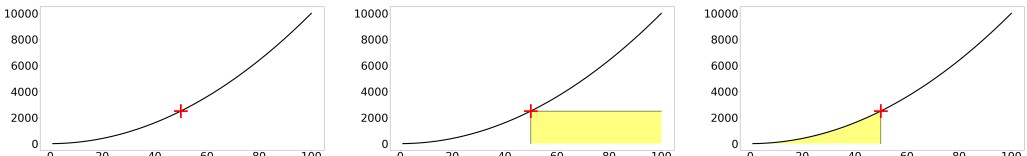

Figure 6: **Left:** illustration of a point (red) $p = (50, 2500)$ on the plot of the 2-log-Lipschitz function (black, see Definition 2.1) $f : [0, \infty) \to \mathbb{R}$, which maps every $x \in [0, \infty)$ to $f(x) = x^2$. **Center:** demonstration of the bound for the points with $x$-value at most $50$, which follows by charging against $f(x)$ utilizing that $r$-log-Lipschitz functions are non decreasing . **Right:** demonstration of the bound for the points with $x$-value larger than $50$, which due to the function $f$ being 2-log-Lipschitz can be bounded using Proposition C.1.

## D    SENSITIVITY BOUND FOR THE MINIMUM OVER SYMMETRIC-$r$ FUNCTIONS

In this section we utilize methods from the previous section to bound the sensitivity of the minimum over symmetric-$r$ functions; see Definition 2.2.

Note that as can be expected by previous works, for example Feldman & Schulman (2012), the coreset size depends on the number of functions that the minimum is over. However, since the data is evenly spaced (which is not the case in Feldman & Schulman (2012)), we managed to obtain a significantly smaller (and simpler) coreset than in Feldman & Schulman (2012).

To simplify the following lemma we will prove the following proposition.

**Proposition D.1.** *Let $r, a \geq 0$, and let $f : [0, \infty) \to [0, \infty)$ be an $r$-log-Lipschitz function; see Definition 2.1. Let $\tilde{f} : [0, \infty) \to [0, \infty)$ s.t. for every $x \geq 0$ we have $\tilde{f}(x) = f(x + a)$. We have that $\tilde{f}$ is an $r$-log-Lipschitz function.*

*Proof.* For every $x \geq 0$ and any $c \geq 1$ we have that

$$\tilde{f}(x \cdot c) = f(x \cdot c + a) \leq f\big((x + a) \cdot c\big) \leq c^r \cdot f(x + a) = c^r \cdot \tilde{f}(x),$$

where the first equality is by the definition of $\tilde{f}$, the first inequality is since $f$ is non-decreasing, and by the definition of $(a, c)$ we have $a \leq a \cdot c$, the second inequality is by the definition of $f$ as an $r$-log-Lipschitz function, and the second equality is by the definition of $\tilde{f}$. Since $f$ is non-decreasing we have that $\tilde{f}$ is non-decreasing, which combined with the previous result proves the proposition. □

For simpler use of the results for the problem in Definition 2.6 we prove the following lemma before the general sensitivity bound.

**Lemma D.2.** *Let $n, k \geq 1$ be integers, where $n \geq 10k$, and let $r \geq 0$. Let $\tilde{f}$ be a symmetric-$r$ function; see Definition 2.2. For every set $X \subset \{1, \cdots, n\}$ of size at least $n/k$ and any value $\tilde{x} \in \{1, \cdots, n\}$ we have*

$$\frac{\tilde{f}(\tilde{x})}{\sum_{x' \in X} \tilde{f}(x')} \leq \frac{(10k)^{r+1}}{n}. \tag{15}$$

*Proof.* Since $\tilde{f}$ is a symmetric-$r$ function, there is $a \in \mathbb{R}$ and $f$ that is an $r$-log-Lipschitz function, such that for every $x \in \{1, \cdots, n\}$ we have $\tilde{f}(x) = f(|x - a|)$. Let $\tilde{x} \in \{1, \cdots, n\}$ and $x = |\tilde{x} - a|$. We have that

$$f(\tilde{x}) \leq \tilde{f}_i(\tilde{x}) \tag{16}$$
$$= f_i(|\tilde{x} - a|) \tag{17}$$
$$= f_i(x), \tag{18}$$

where Equation 16 is since $f(\tilde{x}) = \min_{f' \in F} f'(\tilde{x})$ and $\tilde{f}_i \in F$, Equation 17 is since for every $x' \in \{1, \cdots, n\}$ we have that $\tilde{f}_i(x) = f_i(|x' - a|)$ (from the definition of $f_i$ and $a$), and Equation 18 is by the definition of $x$ as $= |\tilde{x} - a|$. Let $X_1 = \{\lfloor |x - a| \rfloor \mid x \in X, x \leq a\}, X_2 = \{\lfloor |x - a| \rfloor \mid x \in X, x \geq a\}$, and $X_i$ be the largest set among $X_1 \setminus \{0\}$ and $X_2 \setminus \{0\}$. We have

$$\sum_{x' \in X} \tilde{f}(x') = \sum_{x' \in X} f_i(|x' - a|) \tag{19}$$
$$\geq \sum_{x' \in X_i} f_i(x'), \tag{20}$$

where Equation 19 is since for every $x' \in \{1, \cdots, n\}$ we have that $\tilde{f}_i(x) = f_i(|x' - a|)$ (from the definition of $f_i$ and $a$), Equation 20 is since $f_i$ is non non-decreasing and $X_i \subset \{\lfloor |x - a| \rfloor \mid x \in \tilde{X}_i\}$. Combining Equation 16 to equation 18 with Equations 19 and 20 yields

$$\frac{\tilde{f}(\tilde{x})}{\sum_{x' \in X} \tilde{f}(x')} \leq \frac{f_i(x)}{\sum_{x' \in X_i} f_i(x')}. \tag{21}$$

In the following, we will show that we can consider only the case where $a \in [1, n]$.

If $a > n$, for every $x' \in [1, n]$, we have $|x' - a| = |(x' - n) + (n - a)| = |x' - n| + |n - a|$, follows since $x' - n, n - a \leq 0$. If $a < 1$, for every $x' \in [1, n]$, we have $|x' - a| = |(x' - 1) + (1 - a)| = |x' - 1| + |1 - a|$, follows since $x' - 1, 1 - a \geq 0$. Hence, if $a \notin (1, n)$ there is $a' \in \{1, \cdots, n\}$ such that for every $x' \in \{1, \cdots, n\}$ we have $|x' - a| = |x' - a'| + |a' - a|$. Therefore, by assigning $f = f_i$ in Proposition D.1, there is $f_i'$ that is an $r$-log-Lipschitz function and $a' \in [1, n]$, such that for every $x' \in \{1, \cdots, n\}$ satisfies $f_i(|x - a|) = f_i'(|x' - a'|)$. Hence, from now on, we assume that $a \in [1, n]$.

Since $|X| \geq \lceil n/k \rceil, X \subset \{1, \cdots, n\}, a \in \{1, \cdots, n\}$, by the choice of $X_i$ we have that $X_i \subset \{1, \cdots, n\}$ and $|X_i| \geq \lceil n/(2k) - 1 \rceil \geq \lceil n/(2.5k) \rceil$ (using the assumption that $n \geq 10k$). For every $x' \in X_i$, substituting $f = f_i$ in Proposition C.2 yields

$$f_i(x) \cdot \min\{1, (x'/x)^r\} \leq f_i(x'). \tag{22}$$

Let $X' = \{x' \in X_i \mid x' \geq x\}$. We have (if $X_i = X'$ jump Equation 24 expression to equation 27)

$$\frac{f_i(x)}{\sum\limits_{x' \in X_i} f_i(x')} \leq \frac{f_i(x)}{\sum\limits_{x' \in X_i} \left( f_i(x) \cdot \min\left\{1, (x'/x)^r\right\}\right)} \tag{23}$$

$$= \frac{1}{\sum\limits_{x' \in X_i} \min\left\{1, (x'/x)^r\right\}} \tag{24}$$

$$= \frac{1}{|X'| + \dfrac{1}{x^r} \cdot \sum\limits_{x' \in X_i \setminus X'} (x')^r} \tag{25}$$

$$\leq \min\left\{\frac{1}{|X'|}, \frac{x^r}{\sum\limits_{x' \in X_i \setminus X'} (x')^r}\right\}, \tag{26}$$

where Equation 23 is by equation 22, Equation 24 is by dividing both sides by $f_i(x)$, Equation 25 and Equation 26 are by rearranging the expression.

**Case 1, if $|X'| \geq |X_i|/2$:**
By Equations 23,24 and that $|X_i| \geq n/(2.5k)$ we have

$$\frac{f_i(x)}{\sum\limits_{x' \in X_i} f_i(x')} \leq \frac{1}{|X'|} \leq \frac{2}{|X_i|} \leq \frac{5k}{n}. \tag{27}$$

**Case 2, if $|X'| < |X_i|/2$:**
We have $|X_i \setminus X'| \geq |X_i|/2 \geq n/(5k)$; since $|X'| \leq |X_i|/2$ and $|X_i| \geq n/(2.5k)$. Since $(a, \tilde{x}) \in [1, n]^2$, we have $x = |\tilde{x} - a| \leq n$; by the definitions of $a, \tilde{x}$, and $x$. Then, we have

$$\frac{f_i(x)}{\sum\limits_{x' \in X_i} f_i(x')} \leq \frac{x^r}{\sum\limits_{x' \in X_i \setminus X'} (x')^r} \tag{28}$$

$$\leq \frac{x^r}{\sum\limits_{x'=1}^{\lceil n/(5k) \rceil} (x')^r} \tag{29}$$

$$\leq \frac{x^r}{\left(n/(10k)\right)^{r+1}} \tag{30}$$

$$\leq \frac{(10k)^{r+1}}{n}, \tag{31}$$

where Equation 28 is by Equations 23 to 26. Equation 29 is since $|X_i \setminus X'| \geq n/(5k), (X_i \setminus X') \subset [n]$ and $f'(x') = (x')^r$ is an increasing function for $x \geq 0$, Equation 30 is from plugging $n = n/(5k)$ in Proposition C.1, and Equation 31 is by assigning that $x \leq n$ and rearranging.

Combining Equation 21, Equation 27, and Equation 28 to Equation 31 proves the lemma. $\square$

In the following theorem, we provide the desired sensitivity bound.

**Theorem D.3.** *Let $n, k \geq 1$ be integers, such that $n \geq 10k$, and let $r \geq 0$. Let $F \subset \{f : \{1, \cdots, n\} \to (0, \infty)\}$, be a set of $|F| = k$ symmetric-r functions; see Definition 2.2. For every $x \in \{1, \cdots, n\}$, let $f(x) = \min\limits_{f' \in F} f'(x)$. Then, for every $x \in \{1, \cdots, n\}$ we have*

$$\frac{f(x)}{\sum\limits_{x'=1}^{n} f(x')} \leq \frac{(10k)^{r+1}}{n}.$$

*Proof.* By the pigeonhole principle, there is $f_i \in F$ and a set $X_i \subset \{1, \cdots, n\}$ of size at least $n/k$ such that for every $x \in X_i$ we have $f(x) = f_i(x)$; i.e., $f_i$ satisfies the set $X_i$. For every $x \in \{1, \cdots, n\}$ we have

$$\frac{f(x)}{\sum\limits_{x'=1}^{n} f(x')} \leq \frac{f_i(x)}{\sum\limits_{x'=1}^{n} f(x')} \tag{32}$$

$$\leq \frac{f_i(x)}{\sum\limits_{x' \in X_i} f(x')} \tag{33}$$

$$= \frac{f_i(x)}{\sum\limits_{x' \in X_i} f_i(x')} \tag{34}$$

$$\leq \frac{(10k)^{r+1}}{n}, \tag{35}$$

where Equation 32 is since $f(x) = \min\limits_{f' \in F} f'(x)$ and $f_i \in F$, Equation 33 is since $X_i \subset \{1, \cdots, n\}$, Equation 34 is from the definition of $X_i$ as a set such that for every $x \in X_i$ we have $f(x) = f_i(x)$, and Equation 35 is by plugging the result of Lemma D.2. □

## E   DETERMINISTIC CORESET CONSTRUCTION

The following lemma is a modification of Lemma 11 from Rosman et al. (2014), which is significantly influenced by it; we use constant sensitivity and move from sums to integrals.

**Lemma E.1.** *Let $k \geq 1$ and let $f : [0,1] \to [0,\infty)$ be a $k$-piece-wise monotonic function, where $t = \int_0^1 f(x)dx > 0$. Let $s$ such that for every $x \in [0,1]$ we have $f(x) \leq ts$. Put $\epsilon \in (0,1)$ and let $\epsilon' = \frac{1}{\lceil (2ks)/\epsilon \rceil}$. Let $S := \{i \cdot \epsilon' \mid i \in \{0, \cdots, 1/\epsilon'\}\}$. We have that $\left| \frac{1}{|S|} \cdot \sum\limits_{x \in S} f(x) - t \right| \leq \epsilon t$.*

*Proof.* For every $i \in [0,1]$ let $h(i) = f(i)/(st)$. We will prove that

$$\left| \int_0^1 h(x)dx - \frac{1}{|S|} \cdot \sum_{x \in S} h(x) \right| \leq 2\epsilon' k. \tag{36}$$

Multiplying this by $ts$ yields

$$\left| \int_0^1 f(x)dx - \frac{1}{|S|} \cdot \sum_{x \in S} f(x) \right| \leq 2\epsilon' kst = \epsilon t,$$

which proves the lemma.

Since $f$ is $k$-piecewise monotonic, $h$ is $k$-monotonic. Hence, there is a partition $\Pi$ of $[0,1]$ into $k$ consecutive intervals such that $h$ is monotonic over each of these intervals.

For every $j \in S$ let $b(j) = \lceil j/(\epsilon\epsilon' s) \rceil$ and $I_j := \{i \in [0,1] \mid b(i) = b(j)\}$. For every $I \in \Pi$ we define $G(I) = \{j \in S \mid I_j \subset I\}$. Their union is denoted by $G = \bigcup_{I \in \Pi} G(I)$. Hence,

$$\left| \int_0^1 h(x)dx - \frac{1}{|S|} \cdot \sum_{x \in S} h(x) \right| = \left| \sum_{j \in S} \int_{x \in I_j} \Big(h(x) - h(j)\Big)dx \right|$$

$$\leq \left| \sum_{j \in S \setminus G} \int_{x \in I_j} \Big(h(x) - h(j)\Big)dx \right| \tag{37}$$

$$+ \sum_{I \in \Pi} \left| \sum_{j \in G(I)} \int_{x \in I_j} \Big(h(x) - h(j)\Big)dx \right|. \tag{38}$$

We now bound equation 37 and equation 38. Put $j \in B$. By the construction of $S$ in the lemma we have $|I_j \cup S| = 1$ and $\max\{i \in I_j\} - \min\{i \in I_j\} \leq \epsilon'$. Hence,

$$\left| \int_{x \in I_j} \Big(h(x) - h(j)\Big)dx \right| \leq \epsilon' \cdot \left( \max_{x \in I_j} h(i) - \min_{x \in I_j} h(i) \right) \leq \epsilon', \tag{39}$$

where the second inequality holds since $h(i) \leq 1$ for every $i \in [0,1]$ (follows from the definition of $h$ and $s$). Since each set $I \in \Pi$ contains consecutive numbers, we have $|S \setminus G| \leq |\Pi| \leq k$. Using this and equation 39, bounds equation 37 by

$$\left| \sum_{j \in S \setminus G} \int_{x \in I_j} \Big(h(x) - h(j)\Big)dx \right| \leq |S \setminus G| \cdot \epsilon' \leq 2k\epsilon'. \tag{40}$$

Put $I \in \Pi$ and denote the numbers in $G(I)$ by $\{g_1, \cdots, g_\theta\}$. Recall that $h$ is monotonic over $I$. Without loss of generality, assume that $h$ is non-decreasing on $I$. Therefore, summing equation 39 over $G(I)$ yields

$$\left| \sum_{j \in G(I)} \int_{x \in I_j} \Big(h(x) - h(j)\Big)dx \right| \leq \sum_{j=g_1}^{g_\theta} \left| \int_{x \in I_j} \Big(h(x) - h(j)\Big)dx \right| \tag{41}$$

$$\leq \sum_{j=g_1}^{g_\theta} \epsilon' \cdot \left( \max_{x \in I_j} h(i) - \min_{x \in I_j} h(i) \right) \tag{42}$$

$$\leq \epsilon' \cdot \sum_{j=g_1}^{g_\theta - 1} \left( \min_{x \in I_{j-1}} h(i) - \min_{x \in I_j} h(i) \right) \tag{43}$$

$$= \epsilon' \cdot \left( \min_{x \in I_{g_\theta}} h(i) - \min_{x \in I_{g_1}} h(i) \right) \tag{44}$$

$$\leq \epsilon', \tag{45}$$

where Equation 45 is since $h(i) \leq 1$ for every $i \in [0,1]$. Summing over every $I \in \Pi$ bounds equation 38 as,

$$\sum_{I \in \Pi} \left| \sum_{j \in G(I)} \int_{x \in I_j} \Big(h(x) - h(j)\Big)dx \right| \leq |\Pi| \cdot \epsilon \leq k \cdot \epsilon'.$$

Plugging Equations 41 to 45 in Equation 38 and Equation 39 in Equation 37 yields equation 36. $\quad\square$

## F  ANALYSIS OF ALGORITHM 1

Using a similar method to the one in the proof of Lemma D.3 yields the following sensitivity bound.

**Lemma F.1.** *Let $\tilde{\ell} : [0,1] \to \mathbb{R}^d$ be a segment; see Definition 1.1. Let $n \geq 10k$. For every $x \in [0,1]$ and any weighted set $Q := (P, w)$ of size $k$ we have*

$$\frac{D(Q, \tilde{\ell}(x))}{\sum_{i=1}^{n} D(Q, \tilde{\ell}(i/n))} \leq \frac{(20k)^{r+1}}{n}.$$

*Proof.* By the pigeonhole principle, there is $p' \in P$ and a set $X \subset \{i/n \mid i \in \{1, \cdots, n\}\}$ of size at least $n/k$ such that for every $x \in X$ we have $p' \in \underset{p \in P}{\arg\min} \, \mathrm{lip}(w(p) \cdot D(p, \tilde{\ell}(x)))$. Let $u, v \in \mathbb{R}^d$ that defines $\tilde{\ell}$ as in Definition 1.1. Let $\ell : \mathbb{R} \to \mathbb{R}^d$ such that for every $x' \in \mathbb{R}$ we have $\ell(x') = u + v \cdot x'$; i.e., an extension of $\tilde{\ell}$ to a line. Let $\tilde{x} \in \underset{x \in \mathbb{R}}{\arg\min} \, w(p) \cdot D(p', \ell(x))$. Let $\mathrm{lip}_{p'} : [0, \infty) \to [0, \infty)$ where for every $\psi \in [0, \infty)$ we have $\mathrm{lip}_{p'}(\psi) = \mathrm{lip}(w(p') \cdot \psi)$. Since lip is an $r$-log-Lifsitz function (see Section 2.1 and Definition 2.1), it holds that $\mathrm{lip}_{p'}$ is also an $r$-log-Lifsitz function. For every $x \in \{1, \cdots, n\}$ we have

$$\min_{p \in P} \mathrm{lip}(w(p) \cdot D(p, \ell(x))) \leq \mathrm{lip}_{p'}(D(p', \ell(x))) \tag{46}$$

$$\leq \mathrm{lip}_{p'}(D(p', \tilde{x}) + D(\ell(x), \ell(\tilde{x}))) \tag{47}$$

$$\leq 2^r \cdot \mathrm{lip}_{p'}(D(p', \ell(\tilde{x})) + 2^r \cdot \mathrm{lip}_{p'}(D(\ell(x), \ell(\tilde{x}))), \tag{48}$$

where Equation 46 is since the minimum is over $P$ and we have $p' \in P$, Equation 47 is by euclidean distance properties, the Equation 48 is by property (2.3) of Lemma 2.1 in Feldman & Schulman (2012), where substituting $M := \mathbb{R}^d$, $r := r$, $\mathrm{dist}(p, q) := D(p, q)$ for every $p, q \in \mathbb{R}^d$, and $D(x) := \mathrm{lip}_{p'}(x)$ for every $x \in [0, \infty)$.

For every $x' \in X$, by looking on the right angle triangle defined by $\ell(x'), \ell(\tilde{x}), p$, we have

$$D(p', \ell(x')) \geq D(p', \ell(\tilde{x})), D(\ell(x'), \ell(\tilde{x})). \tag{49}$$

Hence,

$$\frac{\min_{p \in P} \mathrm{lip}(w(p) D(p, \ell(x)))}{\sum_{x'=1}^{n} \min_{p \in P} \mathrm{lip}(w(p) D(p, \ell(\tilde{x}/n)))} \leq 2^r \cdot \frac{\mathrm{lip}_{p'}(D(p', \ell(x))) + \mathrm{lip}_{p'}(D(\ell(x), \ell(\tilde{x})))}{\sum_{x' \in X} \mathrm{lip}_{p'}(D(p', \ell(x')))} \tag{50}$$

$$\leq \frac{2^r}{n} + 2^r \cdot \frac{\mathrm{lip}_{p'}(D(\ell(x), \ell(\tilde{x})))}{\sum_{x' \in X} \mathrm{lip}_{p'}(D(\ell(\tilde{x}), \ell(x')))}, \tag{51}$$

where Equation 50 is by Equations 46 to 48, and Equation 51 is by equation 49.

Observe that for every $a, b \in \mathbb{R}$ we have

$$D(\ell(a), \ell(b)) = D(v \cdot a + u, v \cdot b + u) \tag{52}$$

$$= \|v \cdot (a - b)\|_2 \tag{53}$$

$$= \|v\|_2 \cdot |a - b|, \tag{54}$$

where Equation 52 is by recalling the definition of $\ell$ as a function that for every $x \in \mathbb{R}$ returns $u + x \cdot v$, Equation 53 is since $D$ is the euclidean distance function, and Equation 54 is by norm-2 properties.

Let $\tilde{f} : \mathbb{R} \to [0, \infty)$ such that for every $x' \in \mathbb{R}$ we have $\tilde{f}(x') = \mathrm{lip}_{p'}(\|v\|_2 \cdot |x' - \tilde{x}|)$. Since $\mathrm{lip}_{p'}$ is an $r$-log-Lifshitz function we have that $\tilde{f}$ is a symmetric $r$-log-Lifshitz function; see Definition 2.2.

Hence,

$$\frac{\text{lip}_{p'}(D(\ell(x), \ell(\tilde{x})))}{\sum_{x' \in X} \text{lip}_{p'}(D(\ell(\tilde{x}), \ell(x')))} = \frac{\text{lip}_{p'}(\|v\|_2 \cdot |x - \tilde{x}|)}{\sum_{x' \in X} \text{lip}_{p'}(\|v\|_2 \cdot |x' - \tilde{x}|)} \tag{55}$$

$$= \frac{\tilde{f}(x)}{\sum_{x' \in X} \tilde{f}(x')} \tag{56}$$

$$\leq \frac{(10k)^{r+1}}{n}, \tag{57}$$

where Equation 55 is by assigning Equations 52 to 54, Equation 56 is by assigning the definition of the function $\tilde{f}$, and Equation 57 is by substituting $X$ by $X$ and $\tilde{f}$ by $\tilde{f}$ in Lemma D.2; recall that $\tilde{f}$ is an $r$-log-Lifshitz function, and that $|X| \subseteq \{1, \cdots, n\}$ is a set of size at least $n/k$.

Combining Equations 50, 51, 55, 56, and 57 proves the lemma. □

Taking $n$ to infinity and using Riemann integration yields the following lemma.

**Lemma F.2.** *Let $\ell : [0, 1] \to \mathbb{R}^d$ be a segment; see Definition 1.1. For every weighted set $Q$ of size $k$ and any $x' \in [0, 1]$ we have*

$$D(Q, \ell(x')) \leq (20k)^{r+1} \cdot \int_0^1 D(Q, \ell(x)) dx.$$

*Proof.* From the definition of Riemann integral, for every $x' \in [0, 1]$ we have

$$D(Q, \ell(x')) \leq (20k)^{r+1} \cdot \lim_{n \in \mathbb{Z}, n \to \infty} \frac{1}{n} \cdot \sum_{i=1}^n D(Q, \ell(x/n)) = (20k)^{r+1} \cdot \int_0^1 D(Q, \ell(x)) dx,$$

where the inequality is by assigning $\ell := \ell$ and $n \to \infty$ in Lemma F.1. □

The assignment of this sensitivity bound in Lemma E.1 requires a bound on the number of monotonic functions, which is presented in the following observation.

**Lemma F.3.** *Let $\ell : [0, 1] \to [0, \infty)$ be a segment; see Definition 1.1. Let $(C, w)$ be weighted set of size $k$ and $f : [0, 1] \to [0, \infty)$ such that for every $x \in [0, 1]$ we have $f(x) = D((C, w), \ell(x))$. It holds that $f$ is $2k$-piece-wise monotonic.*

*Proof.* Observer that the weighted Voronoi diagram for the weighted set $(C, w)$ (using $D(p, p')$, the Euclidean distance, as the distance between any two points $p, p \in \mathbb{R}^d$) has $k$ convex cells, and let $C'$ be the set of those cells. By the definition of the Voronoi diagram, for each cell in $C'$, that corresponds to the point $p \in C$, and for every $x \in [0, 1]$, such that $\ell(x)$ is in the cell, we have

$$\min_{p' \in P} w(p) \cdot D(p', \ell(x)) = w(p) \cdot D(p, \ell(x)).$$

Let $g : [0, 1] \to [0, \infty)$ such that for every $x \in [0, 1]$ we have

$$g(x) = \min_{p' \in P} w(p') \cdot D(p', \ell(x)), \tag{58}$$

i.e., $f(x)$ without applying the $r$-Lifshitz function lip over the euclidean distance. We have that $g$ has an extremum at most ones inside a cell in $C'$ (the single local minimum) and besides this only when "switching" between cells in $C'$, which due to the cells being convex shapes happens at most $k - 1$ times. Therefore, $g$ has at most $2k - 1$ extrema, and as such is $2k$-piece-wise monotonic.

Let $x \in [0, 1]$, and let

$$p \in \arg\min_{p' \in P} w(p') \cdot D(p', \ell(x)).$$

Hence, for every $p' \in P$ it holds that

$$w(p) \cdot D(p, \ell(x)) \leq w(p') \cdot D(p', \ell(x)).$$

Therefore, since $\mathrm{lip} : \mathbb{R} \to \mathbb{R}$ is an $r$-Lifshitz and as such non-decreasing, for every $p' \in P$ we have

$$\mathrm{lip}\big(w(p) \cdot D(p, \ell(x))\big) \leq \mathrm{lip}\big(w(p') \cdot D(p', \ell(x))\big). \tag{59}$$

Combining this with the definition of the functions $f$ and $g$ yields

$$f(x) = \min_{p' \in P} \mathrm{lip}\big(w(p') \cdot D(p', \ell(x))\big) \tag{60}$$

$$= \mathrm{lip}\big(w(p) \cdot D(p, \ell(x))\big) \tag{61}$$

$$= \mathrm{lip}\big(\min_{p' \in P} w(p) \cdot D(p', \ell(x))\big) \tag{62}$$

$$= \mathrm{lip}(g(x)), \tag{63}$$

where Equation 60 is by the definition of $f$, Equation 61 is by equation 59, Equation 62 is by assigning that $\mathrm{lip} : \mathbb{R} \to \mathbb{R}$ is an $r$-Lifshitz and as such non-decreasing, and Equation 63 is by plugging the definition of $g : [0, 1] \to [0, \infty)$ from Equation 58.

Hence, for every $x \in [0, 1]$ we have

$$f(x) = \mathrm{lip}(g(x)).$$

Since $g$ is a $2k$-piece-wise monotonic, and due to $lip$ being a non-decreasing function, we have that $f$ is a $2k$-piece-wise monotonic function; applying $\mathrm{lip}$ over any increasing segment of $g$ would keep the segment increasing, and the same would hold for decreasing segments. $\qquad\square$

**Lemma F.4.** *Let $\ell : [0, \infty) \to (0, \infty)$ be a segment; see Definition 1.1. Put $\epsilon \in (0, 1/10]$. Let $S \subset \mathbb{R}^d$ be the output of a call to* SEG-CORESET$(\ell, \epsilon)$; *see Algorithm 1. Let $w : S \to [0, \infty)$ be a weight function of $S$ such that for every $s \in S$ we have $w(s) = 1/\lceil 4k \cdot (20k)^{r+1} \rceil$. We have that $(S, w)$ is an $(\epsilon, k)$-coreset for $\{\ell\}$; see Definition 2.7.*

*Proof.* Let $Q$ be a weighted set of size $k$ and $f : [0, 1] \to [0, \infty)$ such that for every $x \in [0, 1]$ we have $f(i) = D\big(Q, \ell(x)\big)$. By Lemma F.1 for every $x' \in [0, 1]$ we have

$$f(x') \leq (20k)^{r+1} \cdot \int_0^1 f(x) dx.$$

If $\int_0^1 f\big(g(x)\big) dx = 0$ the lemma holds from the construction of Algorithm 1, hence, we assume this is not the case. Assigning $s = (20k)^{\tilde{r}+1}, \epsilon, k = 2k$ and $f$, which by Observation F.3 is a $2k$-piece-wise monotonic, in Lemma E.1 combined with the construction of Algorithm 1 proves the lemma. $\qquad\square$

## G  ANALYSIS OF ALGORITHM 2.

Combining Lemma F.4 with Theorem 2.5 yields the following; for the definitions of $r, t$, and $d^*$ see Definition 2.3.

**Theorem G.1.** *Let $L$ be a set of $n$ segments; see Definition 1.1. Put $\epsilon, \delta \in (0, 1/10]$. Let $(S, w)$ be the output of a call to* CORESET$(L, k, \epsilon, \delta)$; *see Algorithm 2. Let $n' = \dfrac{8kn \cdot (20k)^{r+1)}}{\epsilon}$ and $k'$ denote $(k + 1)^{O(k)}$.*
*Then Claims (i)–(iii) hold as follows:*

(i) *With probability at least $1 - \delta$, we have that $(S, w)$ is an $(\epsilon, k)$-coreset for $L$; see Definition 2.7.*

(ii) *The size of the set $S$ is in*

$$\frac{k' \cdot \log(n')^2}{\epsilon^2} \cdot O\left(d^* + \log\left(\frac{1}{\delta}\right)\right).$$

(iii) *The computation time of the call to* CORESET$(L, k, \epsilon, \delta)$ *is in*

$$n'tk' + tk' \log(n') \cdot \log\big(\log(n')/\delta\big)^2 + \frac{k' \log(n')^3}{\epsilon^2} \cdot \left(d^* + \log\left(\frac{1}{\delta}\right)\right).$$

*Proof.* Properties (ii) and (iii) follows from the construction of Algorithm 2, and the corresponding Claims (ii) and (iii) in Theorem 2.5. Hence, we will prove, the only remaining claim, Claim (i) of the theorem.

Let $P' \subset \mathbb{R}^d$ and $\epsilon' > 0$ as computed in the call to CORESET$(L, \epsilon, \delta)$. Let $\psi : P' \to [0, \infty)$ such that for every $p \in P$ we have $\psi(p) = 1/\epsilon'$. By Lemma 2.8 and the construction of Algorithm 2 We have that $(P', \psi)$ is an $(\epsilon/2)$-coreset for $L$; see Definition 2.7. By Theorem 2.5, more specifically its Claim (i), and the construction of Algorithm 2, with probability at least $1 - \delta$, for every $Q$, a weighted set of size $k$, we have

$$\left| \sum_{p \in S} w(p) \cdot D(Q, p) - \frac{1}{\epsilon'} \cdot \sum_{p \in P'} D(Q, p) \right| \leq \frac{\epsilon}{4} \cdot \sum_{p \in S} w(p) \cdot D(Q, p). \tag{64}$$

Suppose this indeed occurs. For every $Q$, a weighted set of size $k$, we have

$$\left| \text{loss}(L, Q) - \sum_{p \in S} w(p) \cdot D(Q, p) \right| \leq \left| \text{loss}(L, Q) - \sum_{p \in P'} \psi(p) \cdot D(Q, p) \right| \tag{65}$$

$$+ \left| \sum_{p \in S} w(p) \cdot D(Q, p) - \sum_{p \in P'} \psi(p) \cdot D(Q, p) \right| \tag{66}$$

$$\leq \frac{\epsilon}{2} \cdot \text{loss}(L, Q) + \frac{\epsilon}{4} \cdot \sum_{p \in P'} \psi(p) \cdot D(Q, p) \tag{67}$$

$$\leq \frac{\epsilon}{2} \cdot \text{loss}(L, Q) + \frac{\epsilon}{4(1 - \epsilon)} \cdot \text{loss}(L, Q) \tag{68}$$

$$< \epsilon \cdot \text{loss}(L, Q), \tag{69}$$

where equation 65–equation 66 is by the triangle inequality, equation 67 is since $(P', \psi)$ is an $(\epsilon/2)$-coreset for $L$ and equation 64, equation 68 is since $(P', \psi)$ is an $(\epsilon/2)$-coreset for $L$, and equation 69 is by assigning that $\epsilon \in (0, 1/10]$, hence, $(1 - \epsilon) > 1/2$. Thus, with probability at least $1 - \delta$, $(S, w)$ is an $\epsilon$-coreset for $L$. □

# H   ANALYSIS OF ALGORITHM 3: CORESET FOR CONVEX SHAPES.

For the self-containment of the work, we state previous work on the sensitivity of functions.

## H.1   SENSITIVITY OF FUNCTIONS

In this section, we state a general sensitivity-based coreset that would require the following definitions.

Let $\mathcal{Q}$ be the union of all the $k$ weighted points $(C, w)$.

**Definition H.1** (query space Feldman et al. (2019)). Let $P \subset \mathbb{R}^d$ be a finite non-empty set. Let $f : P \times \mathcal{Q} \to [0, \infty)$ and loss $: \mathbb{R}^{|P|} \to [0, \infty)$ be a function. The tuple $(P, \mathcal{Q}, f, \text{loss})$ is called a *query space*. For every $q \in \mathcal{Q}$ we define the overall fitting error of $P$ to $q$ by

$$f_{\text{loss}}(P, q) := \text{loss}\left( (f(p, q))_{p \in P} \right) = \text{loss}\left( f(p_1, q), \dots, f(p_{|P|}, q) \right).$$

**Definition H.2** (general-$\epsilon$-coreset Feldman et al. (2019)). Let $(P, \mathcal{Q}, f, \text{loss})$ be a query space as in Definition H.1. For an approximation error $\epsilon > 0$, the pair $S' = (S, u)$ is called an *general-$\epsilon$-coreset* for the query space $(P, \mathcal{Q}, f, \text{loss})$, if $S \subseteq P$, $u : S \to [0, \infty)$, and for every $q \in \mathcal{Q}$ we have

$$(1 - \epsilon) f_{\text{loss}}(P, q) \leq f_{\text{loss}}(S', q) \leq (1 + \epsilon) f_{\text{loss}}(P, q).$$

**Definition H.3** (sensitivity of functions). Let $P \subset \mathbb{R}^d$ be a finite and non-empty set, and let $F \subset \{P \to [0, \infty]\}$ be a possibly infinite set of functions. The *sensitivity* of every point $p \in P$ is

$$S^*_{(P, F)}(p) = \sup_{f \in F} \frac{f(p)}{\sum_{p \in P} f(p)}, \tag{70}$$

where $\sup$ is over every $f \in F$ such that the denominator is positive. The *total sensitivity* given a *sensitivity* is defined to be the sum over these sensitivities, $S_F^*(P) = \sum_{p \in P} S_{(P,F)}^*(p)$. The function $S_{(P,F)} : P \to [0, \infty)$ is a *sensitivity bound* for $S_{(P,F)}^*$, if for every $p \in P$ we have $S_{(P,F)}(p) \geq S_{(P,F)}^*(p)$. The *total sensitivity bound* is then defined to be $S_{(P,F)}(P) = \sum_{p \in P} S_{(P,F)}(p)$.

The following theorem proves that a coreset can be computed by sampling according to the sensitivity of functions. The size of the coreset depends on the total sensitivity and the complexity (VC-dimension) of the query space, as well as the desired error $\epsilon$ and probability $\delta$ of failure.

**Theorem H.4** (coreset construction Feldman et al. (2019)). *Let*

- $P = \{p_1, \cdots, p_n\} \subset \mathbb{R}^d$ *be a finite and non empty set, and* $f : P \times \mathcal{Q} \to [0, \infty)$.

- $F = \{f_1, \ldots, f_n\}$, *where* $f_i(q) = f(p_i, q)$ *for every* $i \in [n]$ *and* $q \in \mathcal{Q}$.

- $d'$ *be the dimension of the range space that is induced by* $\mathcal{Q}$ *and* $F$.

- $s^* : P \to [0, \infty)$ *such that* $s^*(p)$ *is the sensitivity of every* $p \in P$, *after substituting* $P = P$ *and* $F = \{f' : P \to [0, \infty] \mid \forall p \in P, q \in \mathcal{Q} : f'(p) := f(p, q)\}$ *in Definition H.3, and* $s : P \to [0, \infty)$ *be the sensitivity bound of* $s^*$.

- $t = \sum_{p \in P} s(p)$.

- $\epsilon, \delta \in (0, 1)$.

- $c > 0$ *be a universal constant that can be determined from the proof.*

- $\lambda \geq c(t+1)\big(d' \log(t+1) + \log(1/\delta)\big)/\epsilon^2$.

- $w : P \to \{1\}$, *i.e. a function such that for every* $p \in P$ *we have* $w(p) = 1$.

- $(S, u)$ *be the output of a call to* CORESET-FRAMEWORK$(P, w, s, \lambda)$ *(Algorithm 1 in Feldman et al. (2019)).*

*Then, with probability at least* $1 - \delta$, $(S, w)$ *is an general-$\epsilon$-coreset of size* $|S| \leq \lambda$ *for the query space* $(F, \mathcal{Q}, f, \|\cdot\|_1)$; *see Definition H.2.*

## H.2 Returning to the analysis of Algorithm 3

The following theorem proves the desired properties of Algorithm 3 as stated at Theorem B.4.

**Theorem H.5.** *Let* $\mathcal{C}$ *be a set of* $n$ *well-bounded convex shapes; see Definition B.3. Put* $\epsilon, \delta \in (0, 1/10]$. *Let* $(P, w)$ *be the output of a call to* CONVEX-CORESET$(\mathcal{C}, k, \epsilon, \delta)$; *see Algorithm 3. Let* $\lambda$ *as computed at Line 1 at the call to Algorithm 3. Let* $n' = \lambda \cdot n$. *Then there is* $(k' \in k + 1)^{O(k)}$ *such that Claims (i)–(iii) hold as follows:*

(i) *With probability at least* $1 - \delta$ *we have that* $(P, w)$ *is an* $(\epsilon, k)$-*coreset of* $\mathcal{C}$; *see Definition B.2.*

(ii) *The size of the set* $S$ *is in*

$$\frac{k' \cdot \log^2 n'}{\epsilon^2} \cdot O\left(d^* + \log\left(\frac{1}{\delta}\right)\right).$$

(iii) *The pair* $(P, w)$ *can be computed in*

$$n' t k' d^2 + t k' \log(n') \cdot \log^2\big(\log(n')/\delta\big) + \frac{k' \log^3 n'}{\epsilon^2} \cdot \left(d^* + \log\left(\frac{1}{\delta}\right)\right)$$

*expected time.*

*Proof.* Property (ii) follows from the construction of Algorithm 3, and the corresponding Claim (ii) in Theorem 2.5. Hence, we will prove, the only remaining claims, Claim (i) and (iii) of the theorem.

**Proof of Claim (i): correctness.**
Let $S \subset \mathbb{R}^d$ and $\lambda > 0$ as computed in the call to CONVEX-CORESET$(\mathcal{C}, \epsilon, \delta)$. Let $\psi : S \to [0, \infty)$ such that for every $p \in S$ we have $\psi(p) = \lambda$. Let $p \in C$, which is in a segment $\ell : [0, 1] \to \mathbb{R}^d$ that is in $C$, by Lemma F.1, for every weighted set $Q$ of size $k$, any $n \geq 10k$, and all $x \in [0, 1]$ we have

$$\frac{D\big(Q, \ell(x)\big)}{\displaystyle\sum_{i=1}^{n} D\big(Q, \ell(i/n)\big)} \leq \frac{(20k)^{r+1}}{n}.$$

Hence, by taking $n$ to infinity by Riemann integrals we have $D\big(Q, \tilde{\ell}(x)\big) \leq (20k)^{r+1} \int_0^1 D\big(Q, \ell(x)\big) dx$. Therefore, since a convex set can be considered as an infinite union of segments, repeating the claim above $d$ times yields that for every weighted set $Q$ of size $k$ for every $p \in C$ we have

$$D\big(Q, p\big) \leq (20k)^{d(r+1)} \int_{p \in C} D\big(Q, p\big) dV. \tag{71}$$

Let $C \in \mathcal{C}$, with the corresponding bounding box $B_C$. Substituting $\epsilon := \epsilon, \delta := \delta, \lambda := \lambda$, the query space $(P, \mathcal{Q}, F, \|\cdot\|_1)$, where $\mathcal{Q}$ is the union over all the weighted sets of size $k$, $d^*$ the VC-dimension induced by $\mathcal{Q}$ and $F$ from Definition 2.3, the sensitivity bound $\tilde{s} := \dfrac{(20k)^{d(r+1)}}{n'}$ for every sufficiently large $n$ (follows from Equation 71), and the total sensitivity $t = (20k)^{d(r+1)}$ that follows from Equation 71 in Theorem H.4, yields that with probability at least $1 - \delta/(n+1)$, a uniform sample (with appropriate weights) of size $\lambda$, as defined at Line 1 of Algorithm 3, is a convex-$(\epsilon/2)$-coreset for $C$. Thus, by the construction of $S_C$ at $B_C$ it follows that $(S_C, \psi)$ is with probability at least $1 - \delta/(n+1)$ a convex-$(\epsilon/2)$-coreset for $C$.

Suppose that this is indeed the case that for every $C \in \mathcal{C}$ we have that $(S_C, \psi)$ is a convex-$(\epsilon/2)$-coreset for $\{C\}$, which occurs with probability at least $1 - \dfrac{n\delta}{n+1}$. Hence, by its construction at Algorithm 3, $(S, \psi)$ is a convex-$(\epsilon/2)$-coreset for $\mathcal{C}$.

By Theorem 2.5, more specifically its Claim (i), and the construction of Algorithm 3, with probability at least $1 - \delta/(n+1)$, for every $Q$ a weighted set of size $k$, we have

$$\left| \sum_{p \in S} w(p) \cdot D(Q, p) - \lambda \cdot \sum_{p \in P'} D(Q, p) \right| \leq \frac{\epsilon}{4} \cdot \sum_{p \in S} w(p) \cdot D(Q, p). \tag{72}$$

Suppose this indeed occurs. For every $Q$, a weighted set of size $k$, we have

$$\left| \text{loss}(\mathcal{C}, Q) - \sum_{p \in S} w(p) \cdot D(Q, p) \right| \leq \left| \text{loss}(\mathcal{C}, Q) - \sum_{p \in P'} \psi(p) \cdot D(Q, p) \right| + \tag{73}$$

$$\left| \sum_{p \in S} w(p) \cdot D(Q, p) - \sum_{p \in P'} \psi(p) \cdot D(Q, p) \right| \tag{74}$$

$$\leq \frac{\epsilon}{2} \cdot \text{loss}(\mathcal{C}, Q) + \frac{\epsilon}{4} \cdot \sum_{p \in P'} \psi(p) \cdot D(Q, p) \tag{75}$$

$$\leq \frac{\epsilon}{2} \cdot \text{loss}(\mathcal{C}, Q) + \frac{\epsilon}{4(1 - \epsilon)} \cdot \text{loss}(\mathcal{C}, Q) \tag{76}$$

$$< \epsilon \cdot \text{loss}(\mathcal{C}, Q), \tag{77}$$

where equation 73–equation 74 is by the triangle inequality, equation 75 is since $(S, \psi)$ is a convex-$(\epsilon/2)$-coreset for $\mathcal{C}$ and equation 72, equation 76 is since $(S, \psi)$ is a convex-$(\epsilon/2)$-coreset for $\mathcal{C}$, and equation 77 is by assigning that $\epsilon \in (0, 1/10]$, hence, $(1 - \epsilon) > 1/2$. Thus, with probability at least $1 - \delta$, $(S, w)$ is an convex-$\epsilon$-coreset for $\mathcal{C}$.

**Proof of Claim (iii): expected running time.**
Since every $C \in \mathcal{C}$ is a convex set, with a given bounding box, the running time of Lines [1–8] of

the call to CONVEX-CORESET$(\mathcal{C}, k, \epsilon, \delta)$ is the order of the number of iterations in the innermost "while" loop at Line 5 of Algorithm 3.

Since every $C \in \mathcal{C}$ is a convex set, by the properties of john ellipsoid it follows that each iteration of the innermost "while" loop, at Line 5 of Algorithm 3, adds with probability at least $1/d^2$ a point to $S_C$; follows from observing that the john ellipsoid $E_C$ bounding $C$ has volume at most $\sqrt{d}$ of $C$ and the bounding box $B_C$ bounds $E_C$ and has a john ellipsoid with volume $d$ times the volume of $E_C$.

Thus, the expected number of iterations in the innermost "while" loop at Line 5 of Algorithm 3 is in $O(nd^2\lambda)$.

Hence, the expected running time of Lines [1–8] of the call to CONVEX-CORESET$(\mathcal{C}, k, \epsilon, \delta)$ is in $O(nd^2\lambda)$. As such, combining this with Claim (iii) of Theorem 2.5 yields the desired expected running time stated in Claim (iii) of the theorem. □

