# OpenReview forum: "Real-time computer vision on low-end boards via clustering motion vectors"
_ICLR.cc/2024/Conference — ICLR 2024 Conference Withdrawn Submission_

### Official Review · Reviewer_aQsA · 2023-10-30

**Soundness:** 1 poor
**Presentation:** 1 poor
**Contribution:** 2 fair
**Rating:** 3
**Confidence:** 4

**Summary:**

The paper proposes a clustering approach based on the idea of coresets. It is demonstrated in the paper that the proposed formulation helps perform tracking and 3D map creation from videos in real time. Few experimental results are shown to demonstrate the claims made in the paper.

**Strengths:**

* Real-time solution to a couple of popular computer vision problems.

**Weaknesses:**

* Not a well-written paper. So many typos and grammatical mistakes.
* The paper widely discusses the existing literature in theory and emphasizes less of the actual contributions of the paper other than making few methods real-time.
* The results are poorly demonstrated. I am unable to conclude how good of a map is obtained using the proposed method.
* Also confusion about video tracing and tracking —see Sec. 3.

Refer Questions section for more comments.

**Questions:**

## Abstract:
* To this end, we utilize motion vectors and clusters. What clusters authors are referring to. I believe it should be clustering algorithms/methods.

* with real-time running time -> that gives real-time performance.

## Introduction
* A meta-survey on such approaches Zou et al. (2019) states that in recent years -> kindly use \citep{} to put parentheses for citation or rewrite this line.

* “fool" -> use `` and ’’ for the apt quotes.


* Figure 1 -> the blue motion vector is hardly visible. Furthermore, kindly use a different color for the blue motion vector as it correlates with the flower in the background.
* Figure 2 -> figures are placed side to side, whereas captions suggest top and bottom. Kindly correct.


* There are many grammatical mistakes in the paper. Kindly improve the writing of the paper.




## General Comment:

With all due respect, tracking and map creation is not computer vision. These are a couple of  problems studied in computer vision. Kindly modify your paper title.

---

### Official Review · Reviewer_qadD · 2023-10-31

**Soundness:** 1 poor
**Presentation:** 1 poor
**Contribution:** 2 fair
**Rating:** 1
**Confidence:** 4

**Summary:**

The paper introduces a fully polynomial randomized approximation scheme for the clustering of motion vectors, which is then applied to the motion vectors produced by standard video encoders to the problem of visual tracking. The approximation scheme is an adaptation of the results of Feldman and Schulman (2012), which is concerned with robust clustering of points in arbitrary metric spaces, to *segments* as defined by the paper. Specifically, points on segments are sampled at uniform intervals, under a condition on the number of points $\epsilon'$ such that the approximation in Feldman and Schulman (2012) is preserved.

**Strengths:**

# Originality
The idea of directly using vector motions produced by video codecs as inputs to computer-vision tasks is interesting, as it is the broader approach of designing FPRAS for computer vision problems.

# Quality
The paper brings a broad review of the literature and is self-contained, including detailed proofs of its several lemmas and theorems.

# Clarity
Every term is defined, and the illustration in Figure 2 helps the reader to understand the geometric meaning of the cost function defined in Equation 1.

# Significance
The paper brings to the attention of the computer-vision community an important class of "probably approximately correct" algorithms, as in the title of Valiant's book.

**Weaknesses:**

# General

The paper is not well organized. The first two sections of the Introduction, titled "Video Tracking" and "Motion Vectors," do not describe the problem addressed by the paper. The subsection "Our Approach" does not describe the approach at all but introduces and illustrates the definition of a cost function which is discussed only much later in the paper. The subsection "Coresets" brings a definition of coreset, followed by an unusually long quote from the paper by Denisov et al. (2023). That section makes a reference to a "segment clustering problem stated in Section 2.1" that has a small coreset, but I was not able to parse the remaining of that paragraph. The references to Jubran et al. (2021) and Rosman et al. (2014) seem unnecessary, as they refer to exceptions (or so I understood) to the stated goal of having coresets which are weighted subsets of the inputs.

Algorithm 1 should be replaced for the simple formula that computes $\epsilon'$. This value is then used to produce samples at uniform intervals on the motion vectors. It is not clear how the claimed novelty of Algorithm 1 generalizes, as stated, previous work by Rosman et al. (2014), which is concerned with fitting segments to points, rather than sampling from segments.

The structure of Algorithm 2 is not at all illustrated by Figure 3, as attempted. A key component of that algorithm (Feldman and Shulman CORESET algorithm (2012)) was replaced in Figure 3 with a different method (Bachem et al. (2018)) for "easier implementation."

# Evaluation
Experimental evaluation is insufficient. There is scant comparison, and no quantitative evaluation other than an unusual computation of frames-per-second (FPS). It is not valid to subtract all computing times but clustering from the pipeline, divide the number of frames by whatever remains and claim that as an FPS.

The role of Artuhr and Vassilvitskii (2007) in the empirical evaluations is unclear since the output of Algorithm 2 should be a clustering of the segments. One the other hand, there is no mention of Algorithm 2 in that section, only of Algorithm 1.

The steps of the video tracking method are unclear. The is no explanation for what "Add for each motion vector its degree to (0, 10 and (1, 0)". The is no discussion of how one moves on beyond $k = 2$.

The jump from clustering of motion vectors to map creation leaves a gaping hole in the paper. The empirical evaluation of 3D map creation follows similar steps, which are repeated almost verbatim and should be omitted.

There are citations that are unusual to the computer-vision community: the OpenCV library, the Python 3 reference Manual, an Ubuntu Linux guide, the Rasberry Pi user guide, Vigdear manual, CutstomTkInter, and others.

The Conclusion section of the paper cannot be moved to an appendix.

**Questions:**

- It is curious that the number of samples on a segment does not depend on the length of the segment, according to Algorithm 1. Is there any intuition for why?

- Still in Algorithm 1, it is correctly stated that $r$ is defined in Definition 2.3; however, given the comment on the second paragraph following the description of the algorithm ("Note that $r$ in Algorithms 1..."), it should be provided as an input, since the function $D$ to which $r$ corresponds is not.

- I assume the word "tracing," which appears twice in Section 3, is at typo, and "tracking" was meant instead?

- Was Algorithm 2 used at all? What is the purpose of using Arthur and Vassilvitskii (2007) if Algorithm 2 already produces a clustering? How is it possible for Arthur and Vassilvitskii (2007) algorithm to have been implemented in Bradski (2000)?

- How is a motion-vector clustering algorithm applied to map creation? Why computations on Raspberry-Pi and utilization of gyroscope contribute to "fair comparison"?

---

### Official Review · Reviewer_wXa9 · 2023-11-01

**Soundness:** 3 good
**Presentation:** 2 fair
**Contribution:** 3 good
**Rating:** 6
**Confidence:** 4

**Summary:**

This paper introduces innovative computer vision techniques that integrate classical machine learning strategies to enhance efficiency and robustness. The authors showcase the practical impact of their clustering method in video tracking and map creation from video, successfully executing it in real-time on micro-computers. The contributions of the paper encompass a novel clustering algorithm for motion vectors, a coreset-based approach that reduces the computational complexity of the clustering algorithm, and the implementation of the clustering algorithm on low-end boards, enabling real-time performance.

**Strengths:**

S1. The paper is well-written and most of the content is quite easy to follow.
S2. The main contribution of this work is significantly interesting by incorporating traditional machine learning techniques in the age of deep learning.
S3. The proposed vector clustering is theoretically sound, I tried my best to examine most of them and did not find obvious errors.
S4. Overall, I have significant concerns regarding the experimental section of the paper. Firstly, the proposed method is only validated in three application scenarios, and the experimental results are not extensively reported or analyzed, neither in the main text nor in the supplementary material.

**Weaknesses:**

W1. The proofs in Section 2 are rather obscure and difficult for readers without relevant background knowledge to comprehend. Additionally, many crucial steps are relegated to the supplementary material, greatly impacting the readability of this paper.
W2. This paper lacks an introduction and discussion of related works, making it challenging for readers unfamiliar with the field to fully understand the contributions of this article.

**Questions:**

Please check the weaknesses listed above.

---

### Official Review · Reviewer_MMZa · 2023-11-09

**Soundness:** 2 fair
**Presentation:** 1 poor
**Contribution:** 2 fair
**Rating:** 3
**Confidence:** 3

**Summary:**

The paper focuses on a tracking algorithm that takes as input motion vectors obtained from standard video encoders.  The main contribution is the leveraging of the notion of coresets applied to segments, obtaining representative clusters and tracking. The bulk of the paper is on the extension of coresets for point sets to segments.  The tracking algorithm result is illustrated in two examples - big buck bunny video example, and 3D map estimation from drone video.

**Strengths:**

The main claim of the paper is in the generalization coreset ideas for points to segments and in the derivation of a tracking algorithm that is computationally efficient.  Certain claims are made about generalization of previous theoretical work (that I am not fully familiar with and cannot comment).

**Weaknesses:**

While I understand the rationale and setup of the problem for translating motion vector inputs as coresets and tracking,  the  results on the two datasets are not convincing.  While the paper talks a lot about how this approach is substantially better in comparison to neural-net based methods, it fails to refer to any of the classic methods in tracking where clustering, robust statistical methods are used.  The paper does refer to a review paper and states that there are over 1000 articles on the subject.  However, if the central aim of the paper is to demonstrate the advancement in tracking algorithms the paper should demonstrate the effectiveness of the algorithm designed by comparing it with at least one alternative (e.g. mean-shift based tracking ,  Comaniciu et al (CVPR 2000)).  I note that the mean-shift based tracker performed in real-time in low computational power settings for given candidate regions in a video over two decades ago.

**Questions:**

I have several questions that will help me identify what the central contributions are and on how the proposed method outperforms over other methods in the state of the art.

1) Is your contribution mainly the extension of coreset idea to segments?   There has been work on coresets for sets of lines (e.g. Coreset for Line-Sets Clustering, Lotan et al, 2022).  Please elaborate on how your method is different.
2) Have you compared your tracker with other methods in opencv and if so, what was the outcome? You refer to OpenCV in your paper and it is not clear from the paper how it was used in your experiments.
3) Can you elaborate on the tradeoff between computational complexity of your technique and (epsilon, delta) choices during coreset construction?

---

### Meta-Review · Area_Chair_Jo5a · 2023-12-07

**Metareview:**

The paper introduces a novel clustering approach grounded in the concept of coresets. The idea of directly using motion vectors generated by video codecs as inputs for computer vision tasks is intriguing. The paper demonstrates the effectiveness of this approach in achieving real-time tracking and 3D map creation from videos. The contributions of the paper include a pioneering clustering algorithm designed for motion vectors, a coresets-based strategy that effectively reduces the computational complexity of the clustering algorithm, and the successful implementation of the clustering algorithm on low-end boards, facilitating real-time performance.

The paper's primary contributions encompass extending generalization coreset concepts from points to segments and the development of a computationally efficient tracking algorithm. The notion of utilizing vector motions directly generated by video codecs as inputs for computer vision tasks is intriguing. The suggested vector clustering approach appears theoretically robust, offering real-time solutions to a couple of computer vision tasks. The paper also brings to the attention of the computer-vision community an important class of "probably approximately correct" algorithms.

All reviews express concerns regarding the insufficient and unconvincing experiments presented in the paper. Firstly, the paper exclusively showcases results in only two scenarios, with the demonstration often lacking clarity, comprehensive reporting, and in-depth analysis. Secondly, a notable gap exists in the absence of comparisons with other methods. Classic tracking methods frequently employ clustering and robust statistical approaches, and it is crucial to benchmark the proposed method against these alternatives to showcase its advancements in the state of the art. The paper's presentation requires enhancement, as highlighted by several reviewers' suggestions. Notably, there is a need for an introduction and a discussion of related work, which would aid readers unfamiliar with the field in comprehending the paper's contributions. The proofs presented in Section 2 are deemed obscure, posing difficulty for readers lacking relevant background knowledge to grasp. Additionally, there are suggestions for improving the presentation of algorithms. There was no rebuttal posted by the authors before the deadline for discussion. As a result, all issues raised remain unresolved.

**Justification For Why Not Higher Score:**

There are serious concerns regarding the experiments and presentation of the paper. There was no rebuttal posted by the authors before the deadline for discussion. As a result, all issues raised remain unresolved.

**Justification For Why Not Lower Score:**

N/A

---

### Decision · Program_Chairs · 2024-01-16

Reject